# Assessing parameter identifiability of a hemodynamics PDE model using spectral surrogates and dimension reduction

Mitchel J. Colebank [1,2]*

**1** Department of Mathematics, University of South Carolina, Columbia, South Carolina, United States of America, **2** Department of Biomedical Engineering, University of South Carolina, Columbia, South Carolina, United States of America

\* mjcolebank@sc.edu

## Abstract

Computational inverse problems for biomedical simulators suffer from limited data and relatively high parameter dimensionality. This often requires sensitivity analysis, where parameters of the model are ranked based on their influence on the specific quantities of interest. This is especially important for simulators used to build medical digital twins, as the amount of data is typically limited. For expensive models, such as blood flow models, emulation is employed to expedite the simulation time. Parameter ranking and fixing using sensitivity analysis are often heuristic, though, and vary with the specific application or simulator used. The present study provides an innovative solution to this problem by leveraging polynomial chaos expansions (PCEs) for both multioutput global sensitivity analysis and formal parameter identifiability. For the former, we use dimension reduction to efficiently quantify time-series sensitivity of a one-dimensional pulmonary hemodynamics model. We consider both Windkessel and Structured Tree boundary conditions. We then use PCEs to construct univariate profile-likelihood confidence intervals and show how changes in experimental design improve identifiability. Our work presents a novel approach to determining parameter identifiability and leverages a common emulation strategy for enabling profile-likelihood analysis in problems governed by partial differential equations.

**Data availability statement:** All relevant data are within the manuscript and its Supporting

## Author summary

The calibration of biophysical models is often ill-posed, with the parameter dimensionality typically larger than available data for parameter inference. In addition, these models often employ parameters that are clinically or experimentally interpretable, hence a unique set of estimated parameters are necessary for interpreting biophysical processes. Sensitivity analysis is a necessary tool for reducing the parameter dimensionality by "fixing" noninfluential parameters, yet choosing the cutoff for parameter fixing is problem

Information files. The source code for
simulations and analyses can be found at
https://github.com/mjcolebank/Colebank_
Identifiability_PDE.

**Funding:** The author(s) received no specific
funding for this work.

**Competing interests:** The authors have
declared that no competing interests exist.

dependent. Identifiability methods like profile-likelihood are computationally expensive,
and have traditionally been reserved for relatively fast simulators. We show that emula-
tion, using polynomial chaos, provides a framework for a two-in-one analysis of model
sensitivity and parameter identifiability. Using a pulmonary hemodynamics simulator,
we show how this framework allows for a more formal analysis of the model and its
parameters. Our approach allows us to examine how different measurement modalities
affect the ability to infer biophysical parameters, and is a step forward in developing
data-specific models for understanding cardiovascular disease.

## Introduction

Computational modeling and simulation is a cornerstone of the digital twin pipeline [1,2],
which is now considered the next scientific frontier in personalized medicine. As documented
extensively in the report from the National Academies of Sciences, Engineering, and Medicine
[3], the development of digital twins requires proper uncertainty quantification for informed
decision making. Given that the mathematical simulators, which are often expensive, need to
be made data-specific by solving an inverse problem, there is an inherent need for methods
that identify which parameters should be prioritized for inference. Biophysical problems typ-
ically prescribe meaning to the parameters themselves, and thus inferred parameters may be
viewed as an additional feature for personalized medicine [1,2].

Global sensitivity analysis includes moment and moment-independent methods [4–6] and
seeks to explain how uncertainties in input parameters correspond to uncertainties in model
outputs. Moment based methods are more commonly employed, and utilize the concepts of
expectation (i.e., the average) and variance to quantify how parameters contribute to the vari-
ance of the output [4]. Moment independent methods assess how the entire probability dis-
tribution function (PDF) of the output(s) are shifted or changed in response to changes in
the model inputs [6]. The most popular moment-based sensitivity measure is variance-based
Sobol' indices [5], which attribute proportions of the variance in output space to parameters
and their interactions. These methods have been extended to time-dependent [7] and mul-
tivariate [8] outputs. In the case of expensive simulators, e.g. blood flow models dictated by
partial differential equations (PDEs), surrogate models (or *emulators*) are used [4,9–11]. Poly-
nomial chaos expansions (PCEs) are a special type of spectral polynomial surrogate that are
designed to be orthogonal with respect to a prior probability measure [4,6], making them effi-
cient for calculating Sobol' indices. Parameters with little to no contribution to the variance
of the system are deemed "non-influential", and can be fixed at *apriori* values while influential
parameters are prioritized for inference. Non-influential parameters are considered "practi-
cally non-identifiable" in that they have little to no impact on the model output and will be
difficult to infer from data given an experimental design [5]. Yet, a robust cutoff for which
parameters are non-influential are unavailable and typically heuristic. Thus, new methods for
determining when parameters are both influential and identifiable are needed.

The profile-likelihood is a powerful identifiability method, which assesses which parame-
ters are identifiable by constructing confidence intervals [2,12,13]. Typically these confidence
intervals are calculated for one parameter at a time, though it is feasible to profile multivari-
ate parameter combinations as well. If confidence intervals are bounded, then the parameter
is considered identifiable, whereas parameters with only one or no finite confidence bounds
are deemed (practically) non-identifiable. This approach has been used in multiple applica-
tions [12,13], yet only several studies have assessed identifiability through this method with
PDE simulators [15–17,20,21]. This is largely due to the computational cost of constructing

profile-likelihood confidence intervals, which require solving multiple optimization problems for the profiled parameter. Hence computational cost for PDE simulators is a bottleneck for using such an approach.

We bridge this knowledge gap by using spectral PCE surrogates to quantify model sensitivity using Sobol' indices, and then test for parameter identifiability using univariate profile-likelihood confidence intervals. We consider a one-dimensional (1D) pulse-wave propagation model of the pulmonary circulation with two different classes of boundary conditions: Windkessel models and Structured Tree models of the distal vasculature [22,23]. We begin by presenting a PCE surrogate methodology that combines dimension reduction, via principal component analysis (PCA), to emulate the time-dependent PDE outputs. We then illustrate how Sobol' indices for both PCA and time-dependent outputs can be calculated using the PCE coefficients [8]. The PCEs, built on the PCA representation of the output, are then used to calculate univariate profile-likelihood confidence intervals for each parameter at a significantly lower computational cost. We consider three different experimental designs for our models, and illustrate how sensitivity and identifiability analyses can be used to probe model parameters and determine which parameters should be fixed given an experimental design. Our results directly apply to the field of computational hemodynamics modeling, but also provide a framework for future analyses of PDE models by combining PCEs, global sensitivity analysis, and identifiability analysis.

## Methods

### Fluid dynamics model

The computational hemodynamics model is a simplified version of the three-dimensional Navier-Stokes equations in cylindrical coordinates. The model simulates pulse-wave propagation throughout a network of blood vessels under the assumption that blood flow is laminar, Newtonian, incompressible, and axially dominant [11,24,25]. In addition, we assume that blood vessels are cylindrical and impermeable. The system constitutes a set of nonlinear, hyperbolic, PDEs, given by the mass conservation and momentum balance equations

$$\frac{\partial A}{\partial t} + \frac{\partial q}{\partial x} = 0, \tag{1}$$

$$\frac{\partial q}{\partial t} + \left(\frac{\gamma + 2}{\gamma + 1}\right)\frac{\partial}{\partial x}\left(\frac{q^2}{A}\right) + \frac{A}{\rho}\frac{\partial p}{\partial x} = -\frac{2\pi\mu(\gamma + 2)}{\rho}\left(\frac{q}{A}\right), \tag{2}$$

respectively. The above equation describes the interactions between blood pressure $p(x,t)$, (mmHg), cross-sectional area $A(x,t)$, (cm$^2$) and volumetric flow rate $q(x,t)$, (mL/s). The inertial and viscous shear stress terms in (2) are derived from the assumed velocity ($u = q/A$, cm/s) profile

$$u(r, x, t) = U(x, t)\frac{\gamma + 2}{\gamma}\left(1 - \left(\frac{r}{R(x, t)}\right)^{\gamma}\right), \tag{3}$$

where $U(x,t)$ (cm/s) is the average axial velocity and $\gamma$ is the power-law exponent. To obtain a relatively flat velocity profile as seen in-vivo, we set $\gamma = 9$ [26]. Finally, we relate pressure and area using the linear stress-strain relationship

$$p(x, t) = \frac{4}{3}\frac{Eh}{r_0}\left(\sqrt{\frac{A}{A_0}} - 1\right), \quad \frac{Eh}{r_0} = k_1 e^{-k_2 r_0} + k_3 \tag{4}$$

where $A_0 = \pi r_0^2$ (cm$^2$) is the reference area, $E$ (mmHg) is the Young's modulus, and $h$ (cm) is the wall thickness. The coefficient $Eh/r_0$ is assumed to follow the above exponential relationship where $k_1$ (mmHg), $k_2$(cm$^{-1}$), and $k_3$ (mmHg) are parameters [23].

We consider a relatively small network in this study, including the main, left, and right pulmonary arteries (MPA, LPA, and RPA, respectively), using the dimensions presented in Qureshi et al. [27]. We use a pulmonary blood flow time-series boundary condition at the inlet of the MPA that is representative of MPA flow magnitude and shape [28]. We enforce continuity of total pressure and conservation of flow at each vascular junction. At the distal end of the large vessels, we enforce one of two boundary conditions, as discussed below and shown in Fig 1. The model is solved using a two-step Lax-Wendroff finite difference scheme [23].

## Windkessel boundary conditions

Windkessel models are three-element electrical circuit analogs [29] which represent the proximal and distal resistances to blood flow as well as a total compliance, represented by $R_p$ (mmHg s/ ml), $R_d$ (mmHg s/ ml), and $C_T$ (ml/mmHg) respectively. This circuit model is mathematically represented by the first order differential equation

$$\frac{dp}{dt} = R_p\left(\frac{dq}{dt}\right) + q\left(\frac{R_p + R_d}{R_p R_d}\right) - \frac{p}{R_d C_T}. \tag{5}$$

The Windkessel models are attached to the two terminal vessels in the network, each with their own unique $R_p, R_d$, and $C_T$ value. Hence the full parameter set for the Windkessel model is

$$\theta_{WK} = \left\{k_1, k_2, k_3, R_{p,1}, R_{p,2}, R_{d,1}, R_{p,2}, C_{T,1}, C_{T,2}\right\} \tag{6}$$

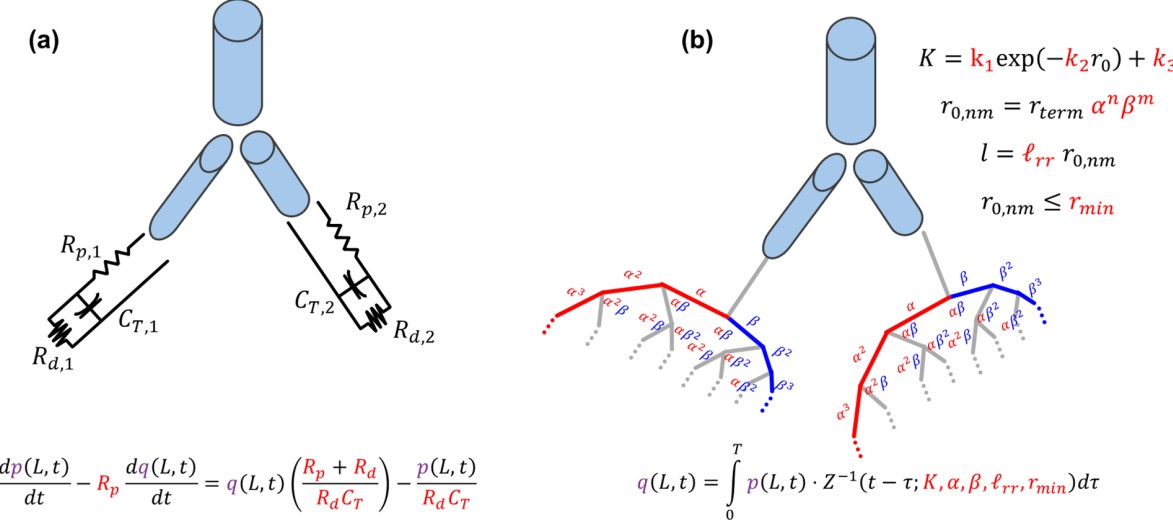

**Fig 1. Schematic of the mathematical models, including three large pulmonary vessels attached to either (a) Windkessel boundary conditions or (b) Structured Tree boundary conditions.** The variables in red denote the parameters to be inferred, as well as their mathematical representations in the boundary conditions. Variables in purple are state variables of the model.

where the subscripts 1 and 2 denote the left and right pulmonary artery boundary conditions, respectively.

## Structured tree boundary conditions

The Structured Tree model [23] is an alternative boundary condition that calculates an asymmetric, bifurcating, synthetic vascular tree at the end of the two terminal vessels. The Structured Tree includes a large radii scaling factor, $\alpha$, a small radii scaling factor, $\beta$, a length-to-radius ratio, $\ell_{rr}$, and a minimum radius for the Structured Tree, $r_{min}$ (cm). Additional details can be found in [23,30]. Whereas the Windkessel model has a unique set of parameters for each terminal branch, we assume a common set of parameters for both the LPA and RPA when using this boundary condition. The coupling between the PDE system and the Structured Tree boundary condition is defined by the convolution integral

$$q(L,t) = \int_0^t p(L,\tau) \cdot Z^{-1}\left(t - \tau; \alpha, \beta, \ell_{rr}, r_{min}\right) d\tau \tag{7}$$

where $x = L$ is the final spatial point in the vessel and $Z(t; \alpha, \beta, \ell_{rr}, r_{min})$ is the total impedance of the Structured Tree, which is calculated under the assumptions of viscous dominant, periodic flow and the parameterization of the Structured Tree geometry itself [23]. Thus, the full parameter set for the Structured Tree model is

$$\boldsymbol{\theta}_{ST} = \{k_1, k_2, k_3, \alpha, \beta, \ell_{rr}, r_{min}\}. \tag{8}$$

## Spectral surrogate

Polynomial chaos expansions (PCEs) are a spectral surrogate modeling technique that uses standard polynomial types that are orthogonal with respect to a prior probability distribution measure [4]. Let $Y = \mathcal{M}(\boldsymbol{\theta})$ denote the output quantity of interest given by the model $\mathcal{M}$ for some $P$ dimensional input parameter vector $\boldsymbol{\theta} \in \mathbb{R}^P$. The PCE approximates the function using the finite summation

$$\mathcal{M}(\boldsymbol{\theta}) \approx \mathcal{M}^{\mathrm{PCE}}(\boldsymbol{\theta}) = \sum_{j=0}^{\mathcal{K}-1} z_j \phi_j(\boldsymbol{\theta}), \quad \phi_j(\boldsymbol{\theta}) = \prod_{i=1}^{P} \psi_{ij}(\theta_i), \tag{9}$$

where $z_j$ are the polynomial coefficients corresponding to the multivariate polynomials $\phi_j(\boldsymbol{\theta})$. These multivariate polynomials are formed by the product of univariate polynomials $\psi_{ij}(\theta_i)$ for polynomial basis function $j$. The polynomial type is associated with the prior distribution for each polynomial; e.g., uniform distributions (used here) correspond to Legendre polynomials [4]. The total number of polynomial basis functions is denoted by $\mathcal{K} = \binom{(P+m)}{m}$, which scales with $P$ parameters and the polynomial order of $m$.

As mentioned above, the PCE polynomials are chosen to be orthogonal with respect to the prior probability measure, giving

$$\mathbf{E}\left[\phi_j(\boldsymbol{\theta}), \phi_{j'}(\boldsymbol{\theta})\right] = \int_\Gamma \phi_j(\boldsymbol{\theta}) \phi_{j'}(\boldsymbol{\theta}) \rho_\theta d\theta = \gamma_j \cdot \delta_{j,j'}, \tag{10}$$

where $\rho_\theta$ is the prior parameter density. The normalization factor $\gamma_j = \mathbf{E}\left[\phi_j^2\right]$ is specific to the polynomial type. We assume uniform priors on all our parameters (after mapping them to the interval $[-1,1]$ [10]), and subsequently use Legendre polynomials, with $\gamma_j = 2/(2j+1)$.

We determine the polynomial coefficients, $Z$ using non-intrusive regression [4]. Given the simulator outputs $Y$, the coefficients of the PCE are calculated using the ordinary least squares solutions

$$Z = \left(\boldsymbol{\Phi}^\top \boldsymbol{\Phi}\right)^{-1} \boldsymbol{\Phi}^\top Y,  \tag{11}$$

where $\boldsymbol{\Phi}$ is the matrix of polynomials evaluated at each parameter value. The size of both the polynomial matrix $\boldsymbol{\Phi}$ and the coefficient matrix $Z$ is dictated by the size of the solution space in $Y$. We use the UQlab software in MATLAB for PCE construction and evaluation [31].

## Dimensionality reduction

The blood flow PDE model includes three dynamic states: pressure, flow, and area. The states are spatiotemporal signals in the three blood vessels. We consider model outputs at the mid-point of each vessel. Rather than calibrate a PCE to these true PDE signals, we emulate on a reduced representation of the multivariate simulator output using PCA [22,32]. Similar to proper orthogonal decomposition (POD) and following from the general decomposition via the Karhunen–Loève expansion [7], PCA constructs a reduced subspace of the original data using a singular value decomposition of the data covariance matrix. As done previously for this model [22], the simulator output can be decomposed via

$$Y(\theta, \mathbf{t}) \approx \boldsymbol{\mu}(\mathbf{t}) + \sum_{q=1}^{M} c_q(\theta, \mathbf{t}) \eta_q(t).  \tag{12}$$

The above decomposition accounts for the time-dependent average, $\mathbf{E}\left[Y(\theta, \mathbf{t})\right] = \boldsymbol{\mu}(\mathbf{t})$, of the model response, as well as the principal component scores, $c_q(\theta)$, and orthonormal basis vectors $\eta_q$. The orthonormal basis vectors provide a rotated coordinate frame for the original model response, and can be used to construct a lower dimensional representation for the output domain, i.e. $M < N_t$, where $N_t$ is the number of time-points in the original signals and $M$ is the selected number of principal components. For $M$, independent principal components, the PCE requires $\boldsymbol{\Phi} \in \mathbb{R}^{\mathcal{K} \times M}$ independent polynomials. The PCE polynomial coefficients are computed by regressing on the PCA scores

$$c_q(\theta) = \sum_{k=0}^{\mathcal{K}-1} z_{qk} \phi_k(\theta), q = 1, \dots, M  \tag{13}$$

for each principal component. The PCE predictions map the parameter vector using the polynomial bases and coefficients defined in Eq (13) to the PCA scores, which can then be transformed into the time-domain using Eq (12). Finally, we transform the PCE based surrogate for the PCA transformed dynamics back to the time-domain when constructing univariate profile-likelihood based confidence intervals. This mapping is denoted by

$$\mathcal{M}^{\mathrm{PCE}}(t; \theta) = \boldsymbol{\mu}(\mathbf{t}) + \sum_{q=1}^{M} \sum_{k=0}^{\mathcal{K}-1} z_{qk} \phi_k(\theta) \eta_q(t)  \tag{14}$$

where $z_{qk}$ is the PCE coefficient corresponding to the $q$th principal component and $k$th polynomial term, $\phi_k(\theta)$ is the $k$th polynomial basis function, and $\eta_q$ is the corresponding principal component basis vector, respectively. We build separate PCE-PCA surrogates, $\mathcal{M}^{\mathrm{PCE}}(t; \theta)$ for each simulator output state used in our analysis, i.e., vessel specific pressure, flow, or area.

## Global sensitivity analysis

The use of spectral-surrogates enables the immediate calculation of variance-based, Sobol' indices once the surrogate is developed. We map the parameter space to the interval $[0,1]$ and denote the scaled parameter space by $\Gamma \in \mathbb{R}^P$. We assume that the parameters of the system are independent in their prior space, which then enables a hierarchical decomposition of the model by

$$\mathcal{M}(\boldsymbol{\theta}) = f_0 + \sum_{i=1}^{P} f_i(\theta_i) + \sum_{1 \le i < j \le P} f_{ij}(\theta_{ij}) + \dots . \tag{15}$$

where $f_i(\theta_i)$ represents the contributions to $\mathcal{M}(\boldsymbol{\theta})$ through the sole effects of $\theta_i$, $f_{ij}(\theta_{ij})$ represents the pairwise contributions, and so on. Under the assumption of independent priors on the parameters, the hierarchical components of the model are orthogonal, and enables a similar hierarchical decomposition of the variance of the system, $V = \mathbf{Var}[Y]$. The partial-variances, $V_i$, attributed to a given parameter $\theta_i$ are then expressed as

$$V_i(\boldsymbol{Y}) = \int_0^1 f_i^2(\theta_i) d\theta_i. \tag{16}$$

The first-order Sobol' index is defined by

$$S_i = \frac{V_i}{V} = \frac{\mathbf{Var}\left[\mathbf{E}\left[Y|\theta_i\right]\right]}{\mathbf{Var}\left[Y\right]}, \tag{17}$$

which represents the proportion of variance attributed to a single parameter, $\theta_i$, while the total-order Sobol' index is defined as

$$S_{T_i} = 1 - \frac{\mathbf{Var}\left[\mathbf{E}\left[Y|\boldsymbol{\theta}_{\sim i}\right]\right]}{\mathbf{Var}\left[Y\right]} \tag{18}$$

where the notation $\boldsymbol{\theta}_{\sim i}$ denotes the set of parameters that does not include $\theta_i$. The total-order index represents the proportion of variance attributed to the parameter $\theta_i$ through all of its interactions with other parameters, hence $S_{T_i} \approx 0$ indicates that $\theta_i$ has little contribution on the variance. This implies that it is functionally non-influential [4,5,8].

We examine both the Sobol' indices for the PCA representation of the output, as well as for the time-dependent output following methods defined by Nagel et al. [8]. For the former, the values of $S_i^q$ and $S_{T_i}^q$, corresponding to the $q$th PCA score sensitivity, are given by

$$S_i^q = \frac{1}{\mathbf{Var}\left[\boldsymbol{Z_q}\right]} \sum_{k \in \mathcal{A}_{S_i}} \left(z_{qk}^2 \gamma_k\right), \quad S_{T_i}^q = \frac{1}{\mathbf{Var}\left[\boldsymbol{Z_q}\right]} \sum_{k \in \mathcal{A}_{S_{T_i}}} \left(z_{qk}^2 \gamma_k\right), \tag{19}$$

where $\mathcal{A}_i$ and $\mathcal{A}_{S_{T_i}}$ represent the set of polynomial coefficients corresponding to only $\theta_i$ and all coefficients that include $\theta_i$, respectively. The variable $\gamma_k$ represents the polynomial normalization factor defined previously. As described in detail by Nagel et al. [8], the coefficients corresponding to the PCE-PCA surrogate can also be used to compute point-wise Sobol' indices describing the original, time-series output. Note that

$$\mathbf{E}\left[Y \mid \theta_i\right] \approx \mathbf{E}\left[\mathcal{M}^{\mathrm{PCE}} \mid \theta_i\right]$$
$$= \boldsymbol{\mu}(\mathbf{t}) + \sum_{q=1}^{M} \mathbf{E}\left[c_q(\boldsymbol{\theta}) \mid \theta_i\right] \eta_q(t)$$

$$= \boldsymbol{\mu}(\mathbf{t}) + \sum_{q=1}^{M} \sum_{k=0}^{\mathcal{K}-1} z_{qk} \mathbf{E}\left[\phi_k(\boldsymbol{\theta}) \mid \theta_i\right] \eta_q \tag{20}$$

and similarly

$$\mathbf{E}\left[Y \mid \theta_{\sim i}\right] \approx \boldsymbol{\mu}(\mathbf{t}) + \sum_{q=1}^{M} \sum_{k=0}^{\mathcal{K}-1} z_{qk} \mathbf{E}\left[\phi_k(\boldsymbol{\theta}) \mid \theta_{\sim i}\right] \eta_q \tag{21}$$

where we remove the time-dependence of $\eta_q$ for ease of notation. We can then write

$$\mathbf{Var}\left[\mathbf{E}\left[Y|\theta_i\right]\right] \approx \mathbf{Var}\left[\mathbf{E}\left[\mathcal{M}^{\mathrm{PCE}}|\theta_i\right]\right]$$

$$= \sum_{q=1}^{M} \mathbf{Var}\left[\mathbf{E}\left[c_q(\boldsymbol{\theta}) \mid \theta_i\right]\right] \eta_q^2 + 2 \sum_{q<q^*} \mathbf{Cov}\left[\mathbf{E}\left[c_q(\boldsymbol{\theta}) \mid \theta_i\right], \mathbf{E}\left[c_{q^*}(\boldsymbol{\theta}) \mid \theta_i\right]\right] \eta_q \eta_{q^*}$$

$$= \sum_{q=1}^{M} S_i^q \mathbf{Var}\left[c_q(\boldsymbol{\theta})\right] \eta_q^2 + 2 \sum_{q<q^*} \mathbf{Cov}\left[\mathbf{E}\left[c_q(\boldsymbol{\theta}) \mid \theta_i\right], \mathbf{E}\left[c_{q^*}(\boldsymbol{\theta}) \mid \theta_i\right]\right] \eta_q \eta_{q^*} \tag{22}$$

where on the last line we rearrange the definition of $S_i^q$ provided in Eq (17). By similar argument, we can rearrange the definition of $S_{T_i}^q$ and write

$$\mathbf{Var}\left[\mathbf{E}\left[Y|\theta_{\sim i}\right]\right] \approx \mathbf{Var}\left[\mathbf{E}\left[\mathcal{M}^{\mathrm{PCE}}|\theta_{\sim i}\right]\right]$$

$$= \sum_{q=1}^{M} \left(1 - S_{T_i}^q\right) \mathbf{Var}\left[c_q(\boldsymbol{\theta})\right] \eta_q^2 + 2 \sum_{q<q^*} \mathbf{Cov}\left[\mathbf{E}\left[c_q(\boldsymbol{\theta}) \mid \theta_{\sim i}\right], \mathbf{E}\left[c_{q^*}(\boldsymbol{\theta}) \mid \theta_{\sim i}\right]\right] \eta_q \eta_{q^*}. \tag{23}$$

The point-wise first and total-order Sobol' indices are then given by

$$S_i^Y = \frac{\displaystyle\sum_{q=1}^{M} S_i^q \mathbf{Var}\left[c_q(\boldsymbol{\theta})\right] \eta_q^2 + 2 \sum_{q<q^*} \mathbf{Cov}\left[\mathbf{E}\left[c_q(\boldsymbol{\theta}) \mid \theta_i\right], \mathbf{E}\left[c_{q^*}(\boldsymbol{\theta}) \mid \theta_i\right]\right] \eta_q \eta_{q^*}}{\mathbf{Var}\left[Y\right]} \tag{24}$$

$$S_{T_i}^Y = 1 - \frac{\displaystyle\sum_{q=1}^{M} \left(1 - S_{T_i}^q\right) \mathbf{Var}\left[c_q(\boldsymbol{\theta})\right] \eta_q^2 + 2 \sum_{q<q^*} \mathbf{Cov}\left[\mathbf{E}\left[c_q(\boldsymbol{\theta}) \mid \theta_{\sim i}\right], \mathbf{E}\left[c_{q^*}(\boldsymbol{\theta}) \mid \theta_{\sim i}\right]\right] \eta_q \eta_{q^*}}{\mathbf{Var}\left[Y\right]}. \tag{25}$$

As noted in the original derivation of this approach by Nagel et al. [8], the orthogonality of the polynomials provides the following covariance definitions:

$$\mathbf{Cov}\left[\mathbf{E}\left[c_q(\boldsymbol{\theta}) \mid \theta_i\right], \mathbf{E}\left[c_{q^*}(\boldsymbol{\theta}) \mid \theta_i\right]\right] = \sum_{k \in \mathcal{A}_{S_i}} z_{qk} z_{q^* k} \tag{26}$$

$$\mathbf{Cov}\left[\mathbf{E}\left[c_q(\boldsymbol{\theta}) \mid \theta_{\sim i}\right], \mathbf{E}\left[c_{q^*}(\boldsymbol{\theta}) \mid \theta_{\sim i}\right]\right] = \sum_{k \in \mathcal{A}_{S_{T_i}}} z_{qk} z_{q^* k} \tag{27}$$

## Profile-likelihood confidence intervals

The profile-likelihood is a frequentist statistical method that explicitly calculates parameter confidence intervals by "profiling" parameters, i.e. by fixing some at specific values an

inferring the other parameters. This can include univariate (a single parameter) or multi-variate based profiling [12]. Here, we focus on constructing univariate parameter confidence intervals. Let $\mathcal{M}(\theta)$ denote the model output and $\mathbf{y}$ be the corresponding, possibly noisy, observations. The negative log-likelihood is then proportional to the weighted sum of square error

$$-LL(\boldsymbol{\theta}) \propto \left(\mathbf{y} - \mathcal{M}(\boldsymbol{\theta})\right)^{\top} \boldsymbol{\Sigma}^{-1}\left(\mathbf{y} - \mathcal{M}(\boldsymbol{\theta})\right). \tag{28}$$

The weight matrix, $\boldsymbol{\Sigma}$, has diagonal entries that represent the variance of the measurement noise and off-diagonal values representing the covariances between observations. Here, we assume noise free, independent measurements and assign the diagonal elements of $\boldsymbol{\Sigma}$ to a scalar value that nondimensionalizes different observations used in the experimental design. Specifically, we consider three experimental designs: (i) only dynamic pressure data in the first branch, (ii) dynamic pressure in the first branch and dynamic area in the daughter branches, and (iii) dynamic pressure in the first branch and dynamic flow in the daughter branches. In these cases, the diagonal elements of $\boldsymbol{\Sigma}$ are the average pressure, flow, or area values of the data.

The univariate profile-likelihood for a parameter $\theta_i$ is then defined as

$$PL_i(\theta_i) = \min_{\boldsymbol{\theta}_{\sim i}}\left[-LL\left(\boldsymbol{\theta}_{\sim i}|\theta_i, \mathbf{y}\right)\right] \tag{29}$$

where $\theta_i$ is the parameter being profiled and $\boldsymbol{\theta}_{\sim i} = \boldsymbol{\theta} \setminus \{\theta_i\}$ is the remaining parameters to be inferred (similar to the definition in Sobol' indices). Univariate confidence intervals can be computed for each parameter by comparing the profile-likelihood to a chi-square distribution

$$\text{CI}(\theta_i) = \{\theta_i| -2\left(PL_i(\theta_i) - LL(\boldsymbol{\theta}_{MLE}|\mathbf{y})\right) \leq \Delta(a)\} \tag{30}$$

where $\boldsymbol{\theta}_{MLE}$ is the maximum likelihood estimator (MLE) of the full parameter set corresponding to the maximum likelihood (minimum of the negative log-likelihood). The difference between the profile-likelihood and MLE evaluated log-likelihood are compared to a chi-squared distribution, $\Delta(a) = \text{icdf}\left(\chi_1^2, 1-a\right)$, with one-degree of freedom and an $1-a$ confidence level. Prototypical univariate profile-likelihood plots can be found in Fig 2. Parameter confidence intervals that are flat suggest non-identifiable parameters, as there are infinite confidence bounds for which the parameter value may take on (Fig 2(a)). This may be a practical or a structural identifiability issue [12,13,20]. A confidence interval that is bounded on one side alone suggests practical identifiability issues, as more data in the design or a reduction in measurement noise may lead to finite confidence bounds on both sides of the MLE (Fig 2(b)). Finally, we consider a parameter practically identifiable if there exist finite confidence bounds around the MLE, providing a typical parabolic negative log-likelihood with finite confidence bounds (Fig 2(c)).

Here, we consider the spectral, PCA surrogate, $\mathcal{M}^{\text{PCE}}(t; \boldsymbol{\theta})$ defined in Eq (14) for calculated the negative log-likelihood. We note the surrogate based negative log-likelihood by $-\widetilde{LL}(\boldsymbol{\theta})$. We consider point-wise confidence intervals at a 95% significance level throughout. As noted previously, we build an independent PCE-PCA surrogate for each output quantity stemming from the PDE simulator. In this work, we are interested in understanding how different experimental designs affect practical identifiability. Thus, using the PCE-PCA surrogate and the profile-likelihood confidence intervals, we assess three different experimental designs, $Di$, for the pulmonary circulation model:

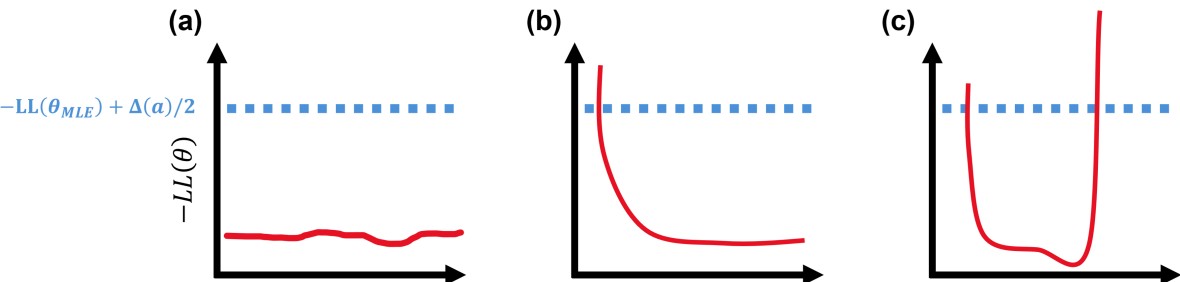

**Fig 2. Example univariate profile-likelihood results and upper bounds for identifiability classification.** An example identifiability cut-off given by the maximum likelihood estimate, $-LL(\theta_{MLE})$, and chi-squared statistics, $\Delta(a)/2$ are provided in blue. A flat profile-likelihood (a) suggests structural or practical identifiability issues, whereas a profile-likelihood that is bounded on one side (b) by the likelihood threshold is considered practically non-identifiable. In contrast, a profile-likelihood that is completely bounded (c) implies a practically identifiable parameter.

1. $D1$: inference using only pressure time-series data in the MPA;
2. $D2$: inference using pressure time-series data in the MPA and time-series area data in the LPA and RPA; and
3. $D3$: inference using pressure time-series data in the MPA and time-series flow data in the LPA and RPA.

We set the measurement noise covariance in Eq (28) to be a diagonal matrix with entries given by the average of the data in the experimental design, i.e. $\mathbf{\Sigma} = \mathrm{diag}\left(\bar{\boldsymbol{y}}_{Di}\right)$, where $\bar{\boldsymbol{y}}_{Di}$ is a vector with the average of each data source included in experimental design $Di$. This scaling of the components in the likelihood is crucial, as the model outputs vary in both units and magnitude, and will bias the profile-likelihood construction if not properly implemented. We note that this definition of $\mathbf{\Sigma}$ is specific to our application, and would instead represent the measurement error variances in a typical multioutput inverse problem [13].

Our profile-likelihood routine is carried out as follows. We start by inferring the full set of parameters, including those that are considered non-identifiable, to replicate the procedure that would be taken with actual measurements. We analyze the results from the initial univariate profile-likelihood across all parameters in combination with global sensitivity results to determine whether there are non-identifiable parameters that may be fixed. We subsequently analyze the univariate profile-likelihood for smaller parameter subsets. This is repeated across the three experimental designs $D1$, $D2$, and $D3$. This reflects a more realistic scenario than fixing parameters at their true value prior to inference, as the true data generating parameters are always unknown. The profile-range for each parameter is calculated as $\pm 50\%$ of the optimal parameter estimate, and thus varies with each experimental design and test dataset. We calculate univariate profile-likelihood confidence intervals by minimizing Eq (28) using the Levenberg–Marquardt algorithm in MATLAB's `lsqnonlin.m` package. We use a finite difference step size of $10^{-4}$ and a maximum of 300 iterations for each profiled parameter value.

## Results

### Surrogate accuracy on test data

We first assess the accuracy of our PCE surrogate on a set of test data for both boundary conditions. We generate 2500 samples to train the Windkessel boundary condition surrogate, of which 2470 successfully run through the simulator without numerical convergence issues. We

use a relatively larger number of samples for the Structured Tree model, given its larger pressure variance (see Fig A in S1 File). We generate 3000 samples for Structured Tree emulation, all of which run successfully. We provide visualizations of several of these PDE solutions for the two boundary condition types in Fig 3. In general, pressure and area dynamics are nearly identical in shape whereas flow waveforms can vary relatively more. Flow signals in the Structured Tree model also tend to vary more than in the Windkessel model. For testing error and later profile-likelihood calculation, we generate 100 test datasets for each boundary condition type. We use a degree five polynomial for the PCE surrogate, which has been shown previously to achieve excellent accuracy in emulating the reduced dimensional output [22]. We use the mean square relative error (MSRE) as a metric for surrogate accuracy, given by

$$\text{MSRE} = \frac{1}{N_{samp}} \sum_{i=1}^{N_{samp}} \left( \frac{Y(\theta^i) - \mathcal{M}^{\text{PCE}}(\theta^i)}{\max(Y(\theta^i))} \right)^2. \tag{31}$$

The model is built on the first five principal components of each vessel's pressure, flow, or area output corresponding to the designs $D1$, $D2$, and $D3$. Table 1 shows the proportion of variance attributed to each component. We use a degree five PCE polynomial for both sets of boundary condition models. As shown in Fig 4, the PCE-PCA accuracy is relatively high in the Windkessel model, with a median MSRE on the order of 2% or less for all five quantities of interest. There is no drastic difference in accuracy, although pressure emulation tends to have the lowest median MSRE in the Windkessel model. In contrast, the Structured Tree model shows relatively larger MSRE values. The pressure MSRE is largest, with a median value of $\approx 2\%$ and outliers at 9% and 15%. Area error metrics are also larger, though the median value is still $\approx 2\%$. The flow errors are relatively similar in magnitude of those from the Windkessel model.

We also compute the relative square error (similar to Eq (31)) attributed to the PCA reconstruction and the emulation of PCA reduced signals. These results, located in Figs B, C, D, and E in S1 File, show that the Windkessel model is substantially easier to approximate by PCA than the Structured Tree. In addition, the error induced by dimension reduction is similar in magnitude to the error from emulation, whereas pressure and area signals tend to have much higher emulation errors relative to reconstruction errors by PCA.

## Global sensitivity analysis

We use Sobol' indices (PCA based and pointwise) to determine how influential parameters are on each output. Fig 5 shows the Sobol' indices across the first three principal components for each boundary condition, with bar graphs and error bars showing the median and range of Sobol' values across the MPA, LPA, and RPA. Results for the Windkessel boundary conditions are provided in Fig 5(a), 5(c), and 5(e). The distal resistances $R_{d,1}$ and $R_{d,2}$ are most influential on the first principal component of pressure, whereas $k_3$ tends to dominate model sensitivity for flow and area. The proximal resistors $R_{p,1}$ and $R_{p,2}$ have relatively small effect on the first principal component of pressure and area, with some variable effects on flow. The exponential stiffness terms $k_1$ and $k_2$ are the least influential parameters for the first principal component, with the compliance parameters being minimally influential as well. The second and third principal components are more sensitive to $k_3$, $R_{p,1}$ and $R_{p,2}$, with the high principal components showing elevated $S_{T_i}^q$ values for previously non-influential parameters. In general, the higher the principal component (and smaller the unexplained variance of the true simulator), the more uniform the $S_{T_i}^q$ values are, with first order indices $S_i$ shrinking in magnitude.

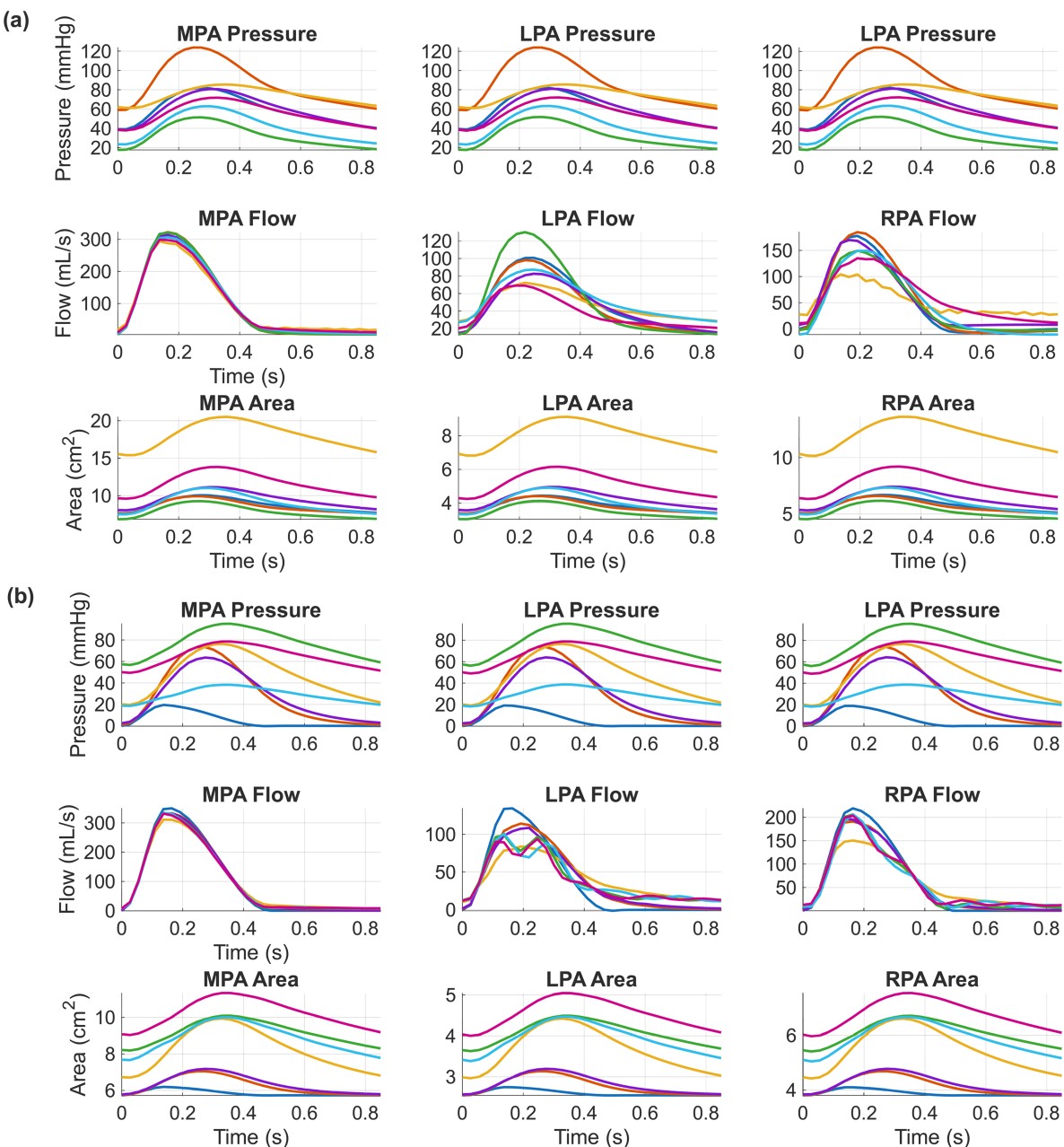

**Fig 3. Seven realizations for the training data set for the 1D model using (a) Windkessel boundary conditions and (b) Structured Tree boundary conditions.** Outputs are provided for each of the signals used in the experimental designs for the profile-likelihood.

We compare these findings to the reconstructed point-wise estimates, $S_i^Y(t)$ and $S_{T_i}^Y(t)$ in Fig 6 built using Eq (24). Results are shown for each vessel. Similar to the findings in Fig 5(a), 5(c), and 5(e), the parameters $R_{d,1}$ and $R_{d,2}$ dominante the pressure sensitivity (they overlap exactly), with $k_3$ showing the next leading influence on the model for both $S_i^Y(t)$ and $S_{T_i}^Y(t)$ metrics across all three branches. The $R_{p,1}$ and $R_{p,2}$ parameters are then the next most influential (again overlapping in magnitude and behavior), in some instances having higher sensitivity metrics than $k_3$. The effects of $k_1$, $k_2$, $C_{T,1}$, and $C_{T,2}$ are negligible in comparison. The

**Table 1. Variance attributed to the first five principal components (PCs).** Right most column represents cumulative total variance from the five principal components (%).

| QoI | PC 1 (%) | PC 2 (%) | PC 3 (%) | PC 4 (%) | PC 5 (%) | Variance (%) |
|---|---|---|---|---|---|---|
| **Windkessel Model** | | | | | | |
| Ves 1 P | 82.2 | 16.4 | 1.4 | 0.0 | 0.0 | 99.9 |
| Ves 2 Q | 58.2 | 34.3 | 7.1 | 0.3 | 0.0 | 99.9 |
| Ves 2 A | 99.9 | 0.1 | 0.0 | 0.0 | 0.0 | 100.0 |
| Ves 3 Q | 58.1 | 33.6 | 6.2 | 0.4 | 0.3 | 98.7 |
| Ves 3 A | 99.9 | 0.1 | 0.0 | 0.0 | 0.0 | 100.0 |
| **Structured Tree Model** | | | | | | |
| Ves 1 P | 88.2 | 9.8 | 1.9 | 0.1 | 0.0 | 99.9 |
| Ves 2 Q | 65.8 | 18.1 | 6.5 | 5.1 | 1.6 | 97.0 |
| Ves 2 A | 97.9 | 1.8 | 0.3 | 0.0 | 0.0 | 100.0 |
| Ves 3 Q | 64.3 | 14.8 | 9.8 | 4.6 | 2.3 | 95.7 |
| Ves 3 A | 97.9 | 1.8 | 0.3 | 0.0 | 0.0 | 100.0 |

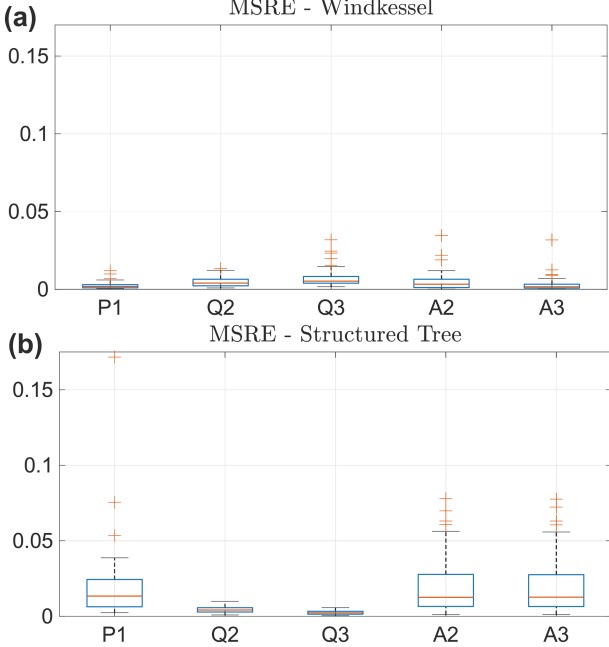

**Fig 4. Surrogate accuracy on 50 test data sets for pressure in the MPA (P1), flows in the LPA and RPA (Q2 and Q3), and areas in the LPA and RPA (A2 and A3).** (a) Windkessel boundary conditions. (b) Structured Tree boundary conditions.

point-wise sensitivity of the flow outputs are distinct across the three branches. The MPA flow output is located downstream from the prescribed flow boundary condition, and thus has a relatively small variance compared to the other two branches. The first-order sensitivities show that $k_3$ is by far the most influential. A similar finding is true for the total-order index, through the other parameters of the system appear to contribute equally. We note that the values of $S_{T_i}^Y(t)$ exceed 1.0 in the MPA, which is attributed to a numerical approximation error in the principal component decomposition (note that this issue does not arise when using four principal components only). For the LPA, we see that flow is most sensitive to $k_3$, $R_{p,1}$ during systole (0.1 - 0.3 seconds) and then most sensitive to $k_3$, $R_{d,1}$, and $R_{d,2}$ during diastole. The total-indices reflect similar trends, with the compliance term $C_{T,1}$ also contributing during

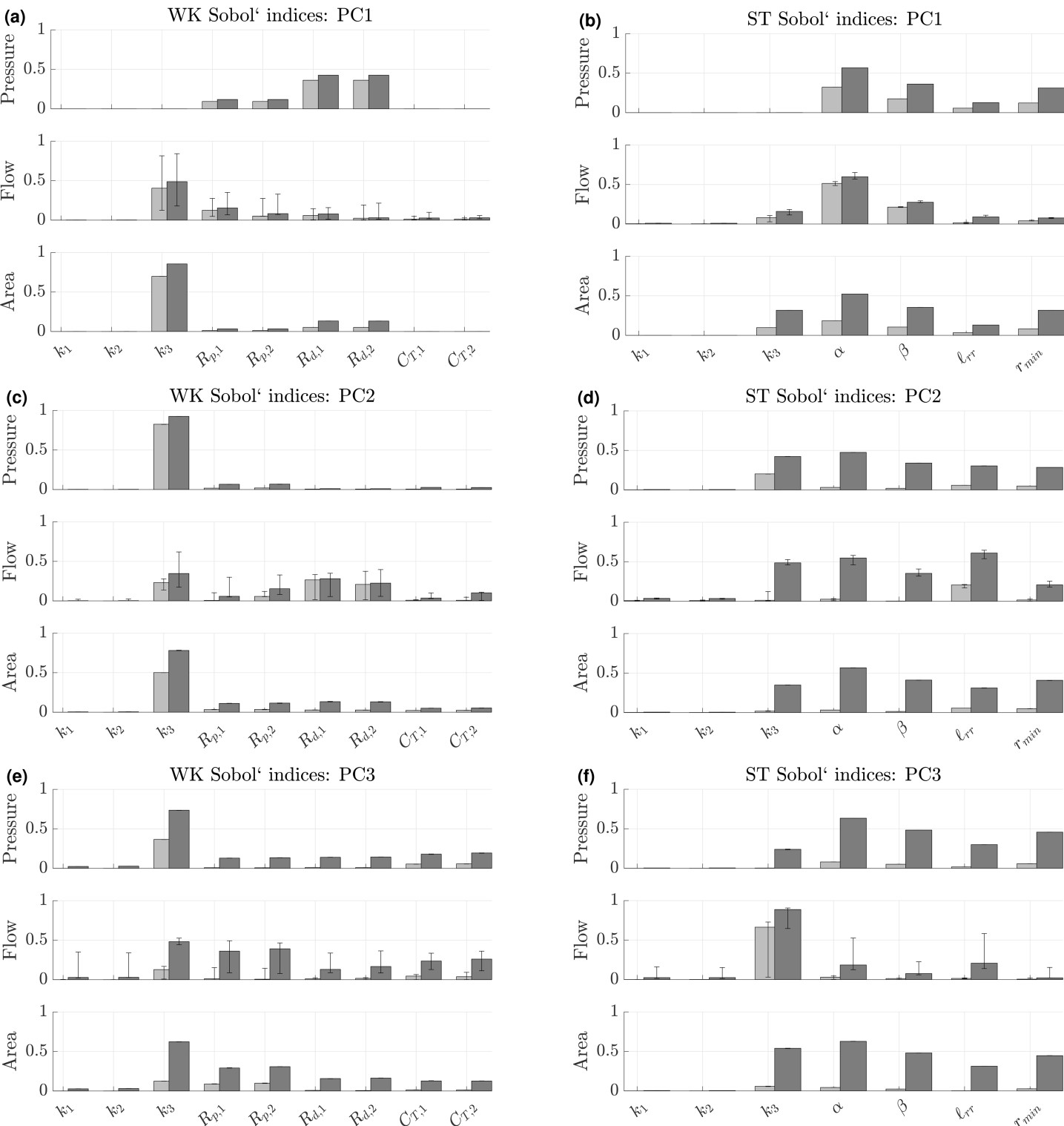

**Fig 5. Sobol' indices for the first three principal components (PCs) using Windkessel boundary conditions (a,c,e) and Structured Tree boundary conditions (b,d,f).** Light bars indicate the median first-order index, $S_i^q$, across the three vessels while dark grey represent the median total-order index. Error bars denote the range for the metrics across the three blood vessels.

 

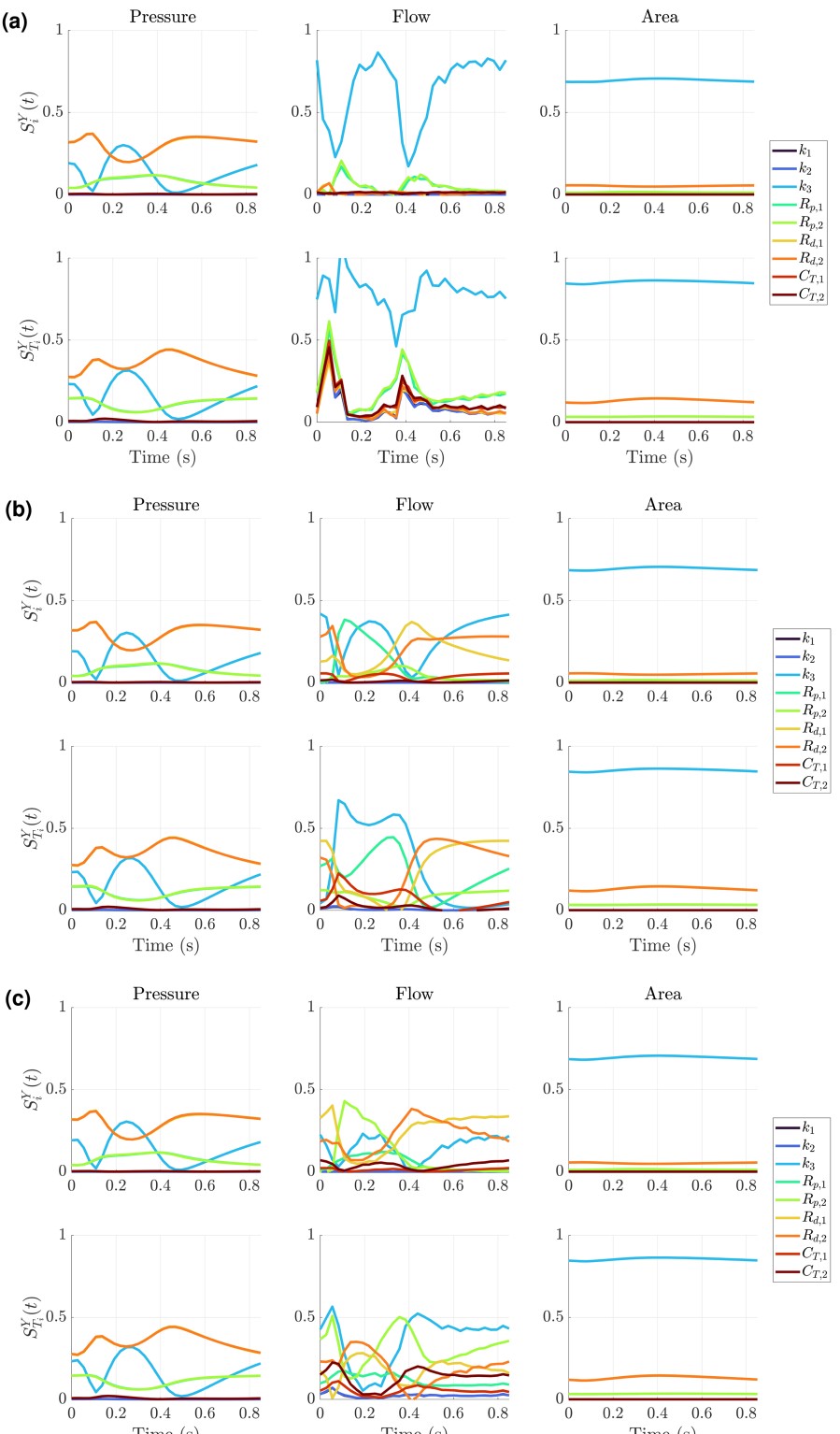

**Fig 6. Point-wise Sobol' indices for the Windkessel boundary conditions calculated from principal component Sobol' metrics $S_i^q$ and $S_{T_i}^q$ via Eq (24).** Results are presented for the (a) MPA, (b) LPA, and (c) RPA in the pulmonary tree.

the upstroke in flow. The RPA flow has elevated sensitivity to $k_3$, $R_{p,2}$, and both distal resistances $R_{d,1}$ and $R_{d,2}$, in terms of both $S_i^Y(t)$ and $S_{T_i}^Y(t)$. Again, the compliance of this branch, $C_{T,2}$ now appears moderately influential as was the case for the LPA. Finally, the area sensitivity is dominated by $k_3$, with only the distal resistance parameters $R_{d,1}$, and $R_{d,2}$ having some effects on area dynamics. This is consistent with the results in Fig 5(a) and 5(c), which shows $k_3$ dominating across the first two principal components for all branches.

The Structured Tree model results in Fig 5(b), 5(d), and 5(f) show similar minimal influence by $k_1$ and $k_2$ across all outputs for the first three principal components as the Windkessel model. Pressure is most sensitive to the Structured Tree parameters $\alpha, \beta, \ell_{rr}$, and $r_{min}$ for the first principal component. The second and third pressure principal components are increasingly more affected by $k_3$. In general $\alpha$ and $\beta$ appear most influential, following by $k_3$, $r_{min}$, and $\ell_{rr}$. The flow results also indicate that the five parameters excluding $k_1$ and $k_2$ dominate model sensitivity. The first through third principal components show that $\alpha$, $\beta$, $k_3$, and $\ell_{rr}$ drive flow sensitivity. There tends to be large variability between branches for the third, fourth, and fifth principal components. Finally, the area predictions are similar in sensitivity to pressure, with the exception that $k_3$ is more influential on area.

We provide the point-wise estimates $S_i^Y(t)$ and $S_{T_i}^Y(t)$ in Fig 7. For the pressure, it is clear that $\alpha$ is most influential, followed by $r_{min}$, $\beta$, and $\ell_{rr}$. The stiffness parameters $k_3$ shows some small changes in model sensitivity for the total order index. The shape of the sensitivity metrics tend to follow the typical time course of a pressure prediction from the model.

For the flow, we see several differences for the three branches. For the MPA (Fig 7(a)), the stiffness $k_3$ dominates during the upstroke in systole, while $\alpha$ and $\beta$ become most influential following this initial increase in flow. The parameter $\ell_{rr}$ becomes most influential during the start of diastole, where $k_3$ and $r_{min}$ also become more influential. During diastole, the parameters $\alpha$ and $\beta$ become most influential. For the LPA flow (Fig 7(b)), $S_i^Y(t)$ suggests that $\alpha$ is most influential for most of the cardiac cycle (except for a brief period where $k_3$ is largest). The parameters $\beta$, $k_3$, $r_{min}$, and $\ell_{rr}$ show larger $S_i^Y(t)$ values in diastole. The curves for $S_{T_i}^Y(t)$ in the LPA show an increase in all parameters, even $k_1$ and $k_2$ during the upstroke in flow. Here, $k_1$ and $k_2$ are still influential even through diastole, though they are smaller than the other parameters. Again we see $\alpha$ having largest influence on the flow on average, with $k_3$ and $\ell_{rr}$ increasing in influence during the start of diastole. Lastly, RPA flow results (Fig 7(c)) show that $k_3$ has larger $S_i^Y(t)$ and $S_{T_i}^Y(t)$ during the upstroke in systole. For $S_{T_i}^Y(t)$ values, we see that $k_1$ and $k_2$ are minimally influential, with only some effects during the start of diastole. The parameter $\ell_{rr}$ appears largely influential during diastole as well.

Finally, the area sensitivity appears nearly identical for all three vessels. In general, $r_{min}$ and $\alpha$ are the most influential, with $\beta$ and $k_3$ being the next most influential. We see a relatively large difference in $S_i^Y(t)$ and $S_{T_i}^Y(t)$ for area (similar to results in Fig 5(b), 5(d), and 5(f)) suggesting higher order interactions between parameters.

## Profile-likelihood: Windkessel model

We use the PCA-PCE spectral surrogate to calculate univariate profile-likelihood confidence intervals for each design described previously. Of the 50 test datasets, we focus on five of these for calculating the profile likelihood. Fig 8 shows three sets of profile-likelihood results for one representative test dataset used for assessing model accuracy in Fig 4. Results for the four other test data sets are provided in the Figs F, G, H, I in S1 File. We first consider inferring all nine parameters (Fig 4(a)) across the three different experimental designs. As expected, the parameters $k_1$ and $k_2$ have nearly flat profile-likelihoods for all three designs, although

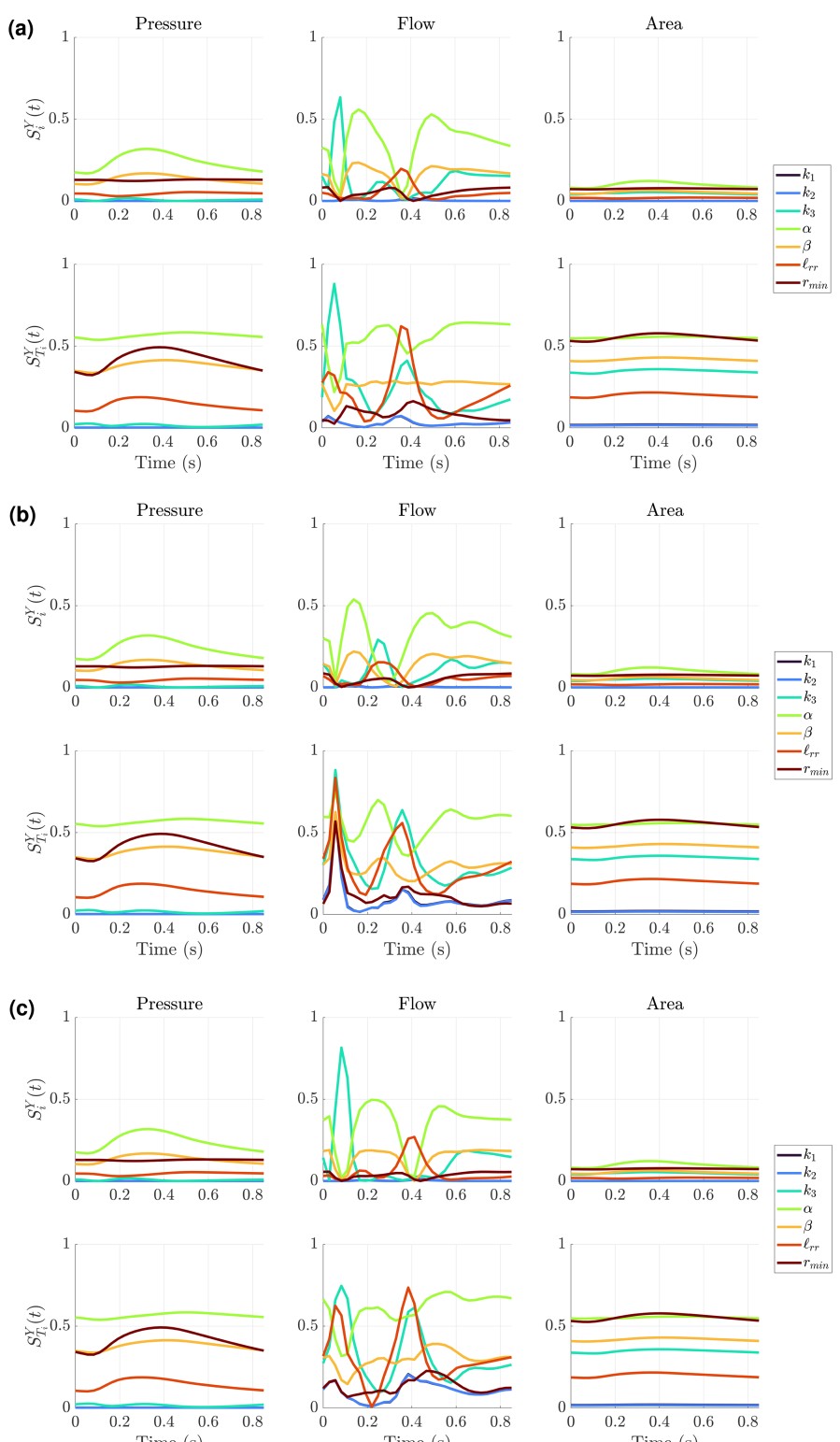

**Fig 7. Point-wise Sobol' indices for the Structured Tree boundary conditions calculated from principal component Sobol' metrics $S_i^q$ and $S_{T_i}^q$ via Eq (24).** Results are presented for the (a) MPA, (b) LPA, and (c) RPA in the pulmonary tree.

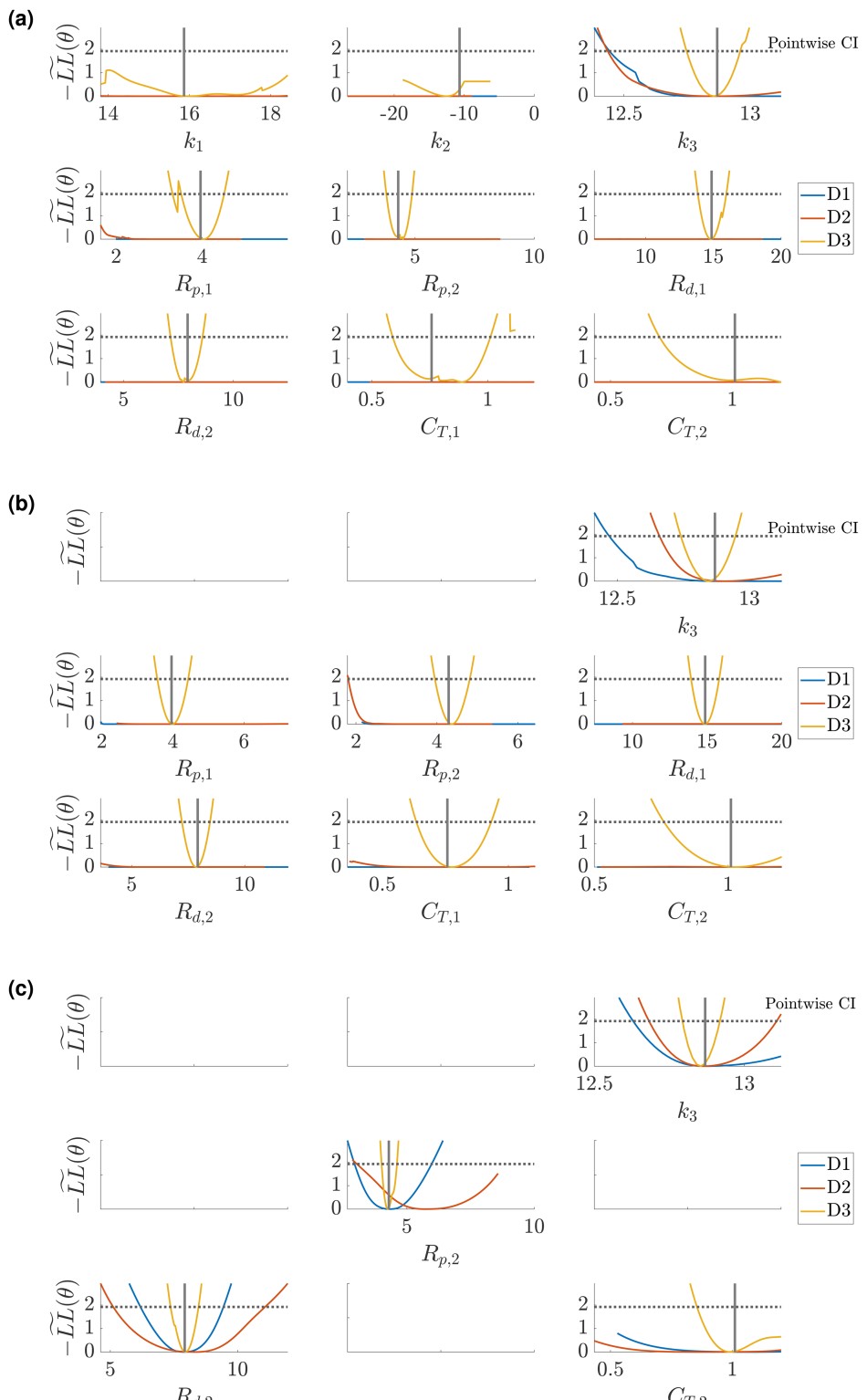

**Fig 8. Profile-likelihood results using the PCA-PCE spectral surrogate.** (a) Univariate profile-likelihood calculated using the three different experimental designs defined previously. The pointwise confidence intervals (defined in Eq (29)) define whether parameters are considered identifiable. (b) A reduced parameter subset where $k_1$ and $k_2$ are not included in the profile-likelihood calculation. (c) A further reduced parameter set where $k_1$ and $k_2$ are fixed, as well as the LPA Windkessel parameters ($R_{p,1}$, $R_{d,1}$, and $C_{T,1}$). Blank plots represent parameters that are fixed.

design $D3$ (MPA pressure and flows in the LPA and RPA) shows some increase in the negative log-likelihood. In general, all nine parameters are not identifiable for designs $D1$ and $D2$ which uses MPA pressure alone and MPA pressure with dynamic area in the LPA and RPA, respectively. The only exception is $k_3$, which has a single finite confidence bound. When using $D3$, parameters $k_3$, $R_{p,1}$, $R_{p,2}$, $R_{d,1}$, $R_{d,2}$, and $C_{T,1}$ all have finite confidence bounds, indicating they are all identifiable. The parameter $C_{T,2}$ is not practically identifiable due to its behavior for increasing values away from the minimum, but may have a finite confidence bound for a larger optimization range.

We subsequently rerun our profile-likelihood analysis with $k_1$ and $k_2$ fixed, given their lack of identifiability and relatively small influence via sensitivity analysis. Results in Fig 4(b) show similar trends to the full parameter set: only $k_3$ has one set of finite confidence bounds for $D1$ and $D2$. This time, $R_{p,2}$ also shows a finite confidence bound for $D2$. Again we see that, when using $D3$, nearly all the parameters have finite confidence bounds except for $C_{T,2}$.

We hypothesize that this finding is related to interactions between the LPA and RPA boundary condition parameters, i.e., left and right boundary conditions may not be jointly identifiable. To test this, we ran an additional experiment where the parameters of the LPA ($R_{p,1}$, $R_{d,1}$, and $C_{T,1}$) were fixed. As shown in Fig 4(c), this improved the identifiability of all four parameters. In particular, $k_3$, $R_{p,2}$ and $R_{d,2}$ are practically identifiable for all three designs ($R_{p,2}$ will be practically identifiable for $D2$ with a larger optimization range). In contrast, $C_{T,2}$ is still not practically identifiable. We performed an additional analysis to further investigate whether fixing both compliance parameters would increase identifiability of the remaining parameters, but still found that only $D3$ provided full parameter identifiability.

We now investigate how univariate profile-likelihood confidence intervals translate to signals in output space. As shown in Fig 9, MPA pressure predictions are provided using parameter values obtained along the profile likelihood in Fig 8(a). To obtain these pressure predictions, we pass the inferred parameter vector obtained from each profiled parameter through the PCE surrogate. Parameter $k_1$ (Fig 9(a)) is not identifiable, and surrogate predictions are indistinguishable along the profile-likelihood. In contrast, $k_3$ (Fig 9(b)) is only slightly practically non-identifiable (unbounded for larger $k_3$ values) for designs $D1$ and $D2$, and identifiable for $D3$. Note that for the latter, surrogate predictions vary greatly, suggesting stronger identifiability of the parameter. Finally, $R_{d,1}$ (Fig 9(c)) is practically non-identifiable for $D1$ and $D2$ (flat profile-likelihood), but is identifiable using $D3$. Model outputs support this claim, overlapping in $D1$ and $D2$, but varying in $D3$. The increase in identifiability (a larger variability in pressure predictions) is related to the change in experimental design, which incorporates different signals for calibration and profile-likelihood construction.

## Profile-likelihood: Structured tree model

We conduct a similar study using the Structured Tree boundary conditions, using five of the 50 testing datasets for calculating the profile-likelihood. Fig 10(a) shows the univariate profile-likelihood results for all seven parameters using a representative test dataset. Results for the four other test data sets are provided in Figs J, K, L, M in S1 File. In contrast to the Windkessel boundary conditions, the likelihood values for the Structured Tree are more sporadic and include multiple local minima and/or possible discrepancies in the surrogate's ability to solve the inverse problem. It is difficult to discern whether any parameters satisfy the definition of identifiability, given the dynamic changes in $-\widetilde{LL}(\theta)$ values. We again consider fixing $k_1$ and $k_2$, as they are the least influential parameters and likely introduce identifiability issues. Results in Fig 10(b) indicate that $k_3$ is now more identifiable, owing to the interactions between $k_1$ and $k_2$ during inference. The parameters $\alpha$, $\beta$, and $\ell_{rr}$ appear identifiable for $D3$,

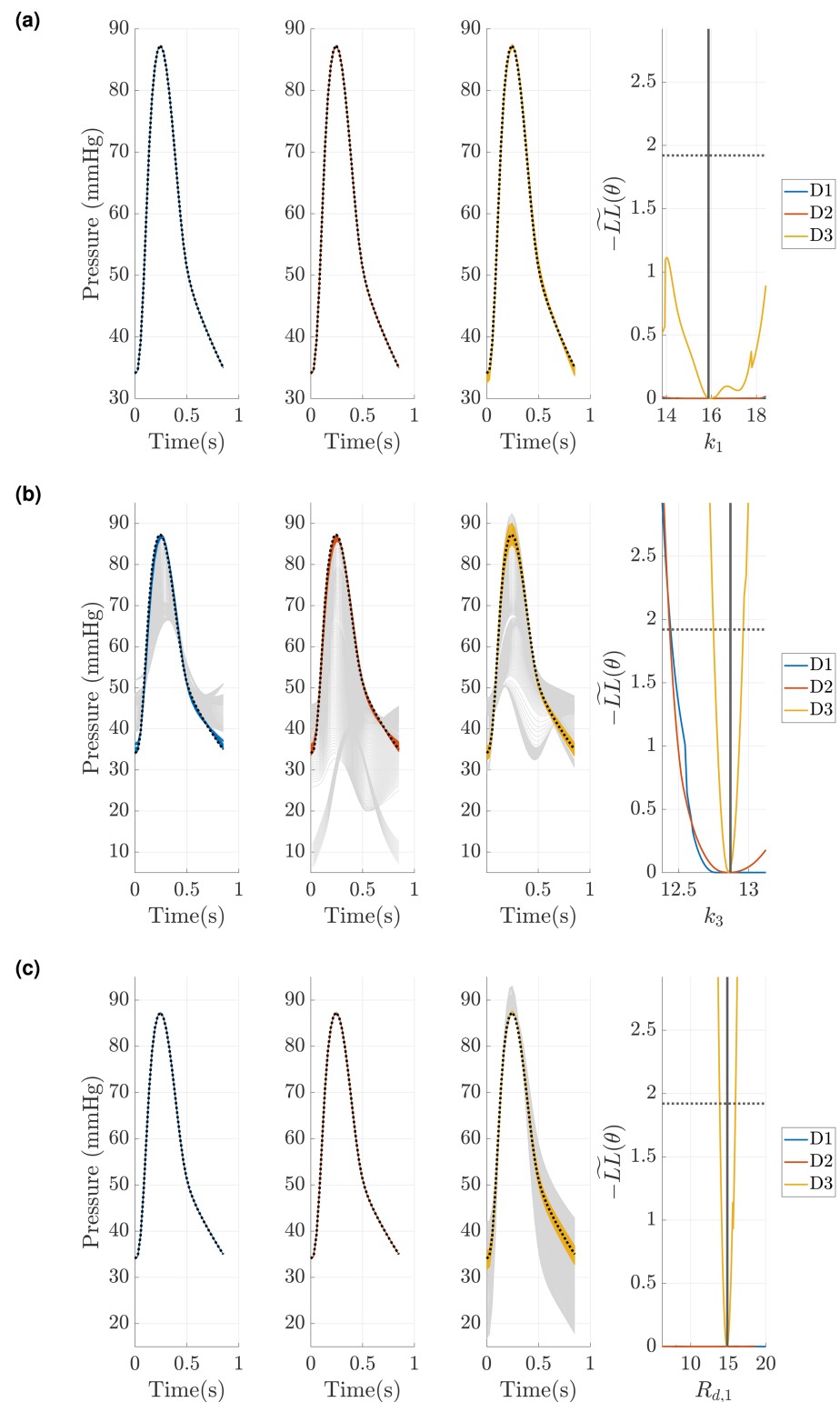

**Fig 9. Model evaluations along the univariate profile-likelihood presented in Fig 8(a) using designs $D1$,$D2$, and $D3$.** Grey predictions represent all model parameters generated during the profile-likelihood, while color-coded predictions are model evaluations within the confidence interval threshold (shown as a dashed horizontal line in the rightmost subplot). The dashed black line represents the true signal used for calibration. (a) Evaluations along the $k_1$ profile-likelihood. (b) Evaluations along the $k_3$ profile-likelihood. (c) Evaluations along the $R_{d,1}$ profile-likelihood.

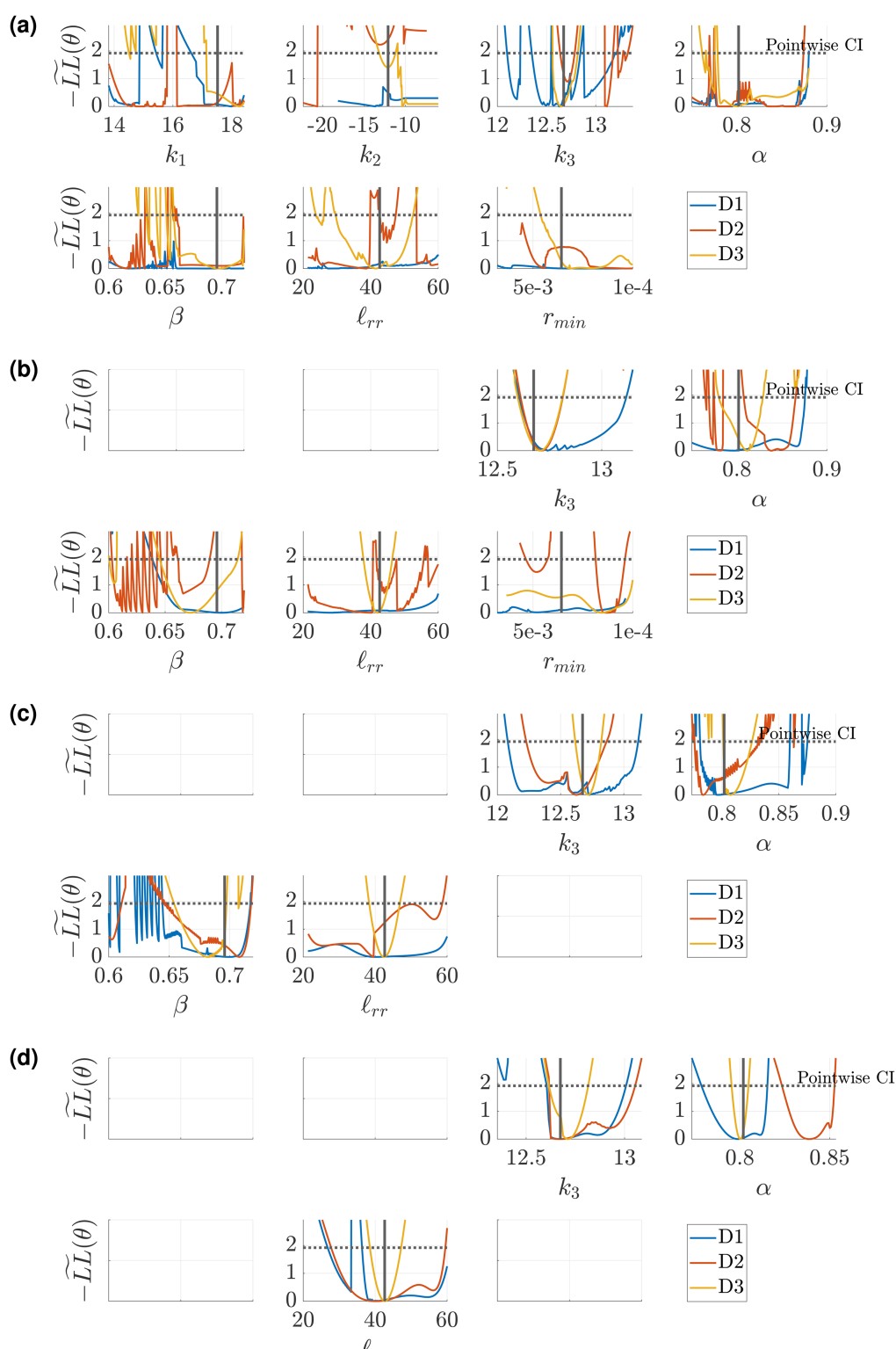

**Fig 10. Profile-likelihood results using the PCA-PCE spectral surrogate.** (a) Univariate profile-likelihood calculated using the three different experimental designs defined previously. The pointwise confidence intervals (defined in Eq (29)) define whether parameters are considered identifiable. (b) A reduced parameter subset where $k_1$ and $k_2$ are not included in the profile-likelihood calculation. (c) A further reduced parameter set where $\beta$ is fixed. (d) Further reduced parameter set with $r_{min}$ fixed. Blank plots represent parameters that are fixed.

but are sporadic with multiple minima for other designs. The parameter $r_{min}$, though influential as shown in Figs 5(b), 5(d), and 5(f) and 7, appears difficult to identify. Hence, we fix $r_{min}$ and again conduct our univariate profile-likelihood analysis. Fig 10(c) shows improved identifiability for $\alpha$, $\beta$, and $\ell_{rr}$ for designs $D2$ and $D3$. As a final analysis, we consider the three dimensional parameter space with $k_3$, $\alpha$, and $\ell_{rr}$, as shown in Fig 10(d). We fix $\beta$ based on its sporadic behavior. These results provide practical identifiability for all three parameters in $D3$, with minimal sporadic likelihood values. The parameter $\alpha$ is incorrectly identified using $D2$ during the initial inference procedure, hence the univariate profile-likelihood confidence bounds are located far away from the results using $D1$ and $D3$.

Similar to Fig 9, we provide output signals that are within the univariate profile-likelihood confidence threshold, as well as signals that are beyond the confidence limits, in Fig 11. We use the four dimensional parameter space presented in Fig 10(c), and provide output signals along the univariate profile-likelihoods for $k_3$, $\alpha$, and $\ell_{rr}$. We see that both $k_3$ and $\alpha$ provide large variability in pressure signals above the confidence interval, whereas $\ell_{rr}$ provides only small deviations in pressure above the confidence threshold. The parameters $k_3$ and $\alpha$ are more influential on pressure than $\ell_{rr}$, hence they have larger deviations in output space.

### Profile-likelihood: Simulator

The full PDE simulator is too expensive to use for profile-likelihood analyses, hence the use of a spectral surrogate. However, we can use the resulting univariate profile-likelihood parameter estimates to assess whether the values of $-\widetilde{LL}(\theta)$ are comparable with the true simulator. Figs 12 and 13 compare the PCA-PCE spectral surrogate values of the profile-likelihood compared with the PDE simulator derived likelihood for the Windkessel and Structured Tree boundary conditions, respectively. Each simulator curve is generated by passing the profile-likelihood parameters to the simulator and then calculating the same weighted likelihood function as used in the profile-likelihood calculation. For the Windkessel model, we investigate the univariate profile-likelihood parameter values for the parameter subset including $k_3$, $R_{p,2}$, $R_{d,2}$, and $C_{T,2}$ (similar to Fig 8(c)). For the Structured Tree model, we investigate the univariate profile-likelihood parameter values for the parameter subset including $k_3$, $\alpha$, and $\ell_{rr}$ (similar to Fig 10(d)).

Though the confidence interval shapes do not match exactly, we observe that the conclusions about identifiability agree between the surrogate framework and the true simulator. Design $D3$ is the most informative design for both models, with $D2$ providing some additional information for parameter identification over $D1$ but not necessarily guaranteeing that parameters are identifiable. As expected, the Windkessel results appear more consistent between the simulator and surrogate (which had a smaller emulation error), whereas the simulator results for the Structured Tree model show how the surrogate and simulator outputs have non-smooth likelihood values. However, the results in Figs 12 and 13 show that the confidence bounds are similar when using the surrogate and simulator.

## Discussion

We use spectral surrogates to determine parameter influence via sensitivity analysis and identifiability by constructing univariate profile-likelihood confidence intervals. While multiple studies have used spectral surrogates in the context of pulse-wave propagation PDE models, none have considered extending these surrogates for formal identifiability analyses. In addition, we present a relatively simple procedure for handling vectorized outputs by reducing the output dimensionality using PCA. We use this framework to assess the sensitivity of different model outputs and test different experimental designs for parameter identifiability.

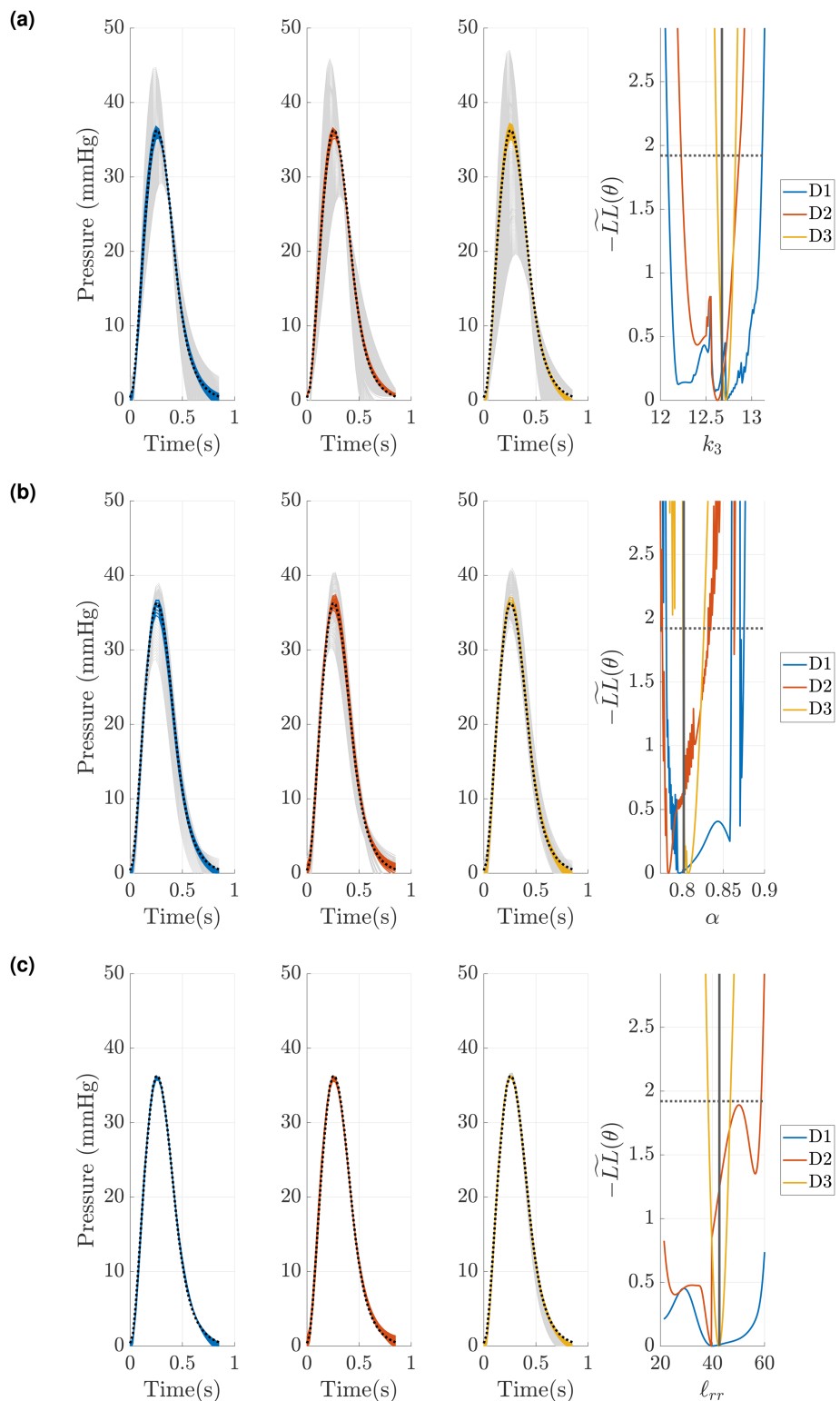

**Fig 11. Model evaluations along the univariate profile-likelihood presented in Fig 10(c) using designs *D*1,*D*2, and *D*3.** Grey predictions represent all model parameters generated during the profile-likelihood, while color-coded predictions are model evaluations below the confidence interval threshold (shown as a dashed horizontal line in the rightmost subplot). The dashed black line represents the true signal used for calibration. (a) Evaluations along the $k_3$ univariate profile-likelihood. (b) Evaluations along the $\alpha$ univariate profile-likelihood. (c) Evaluations along the $\ell_{rr}$ univariate profile-likelihood.

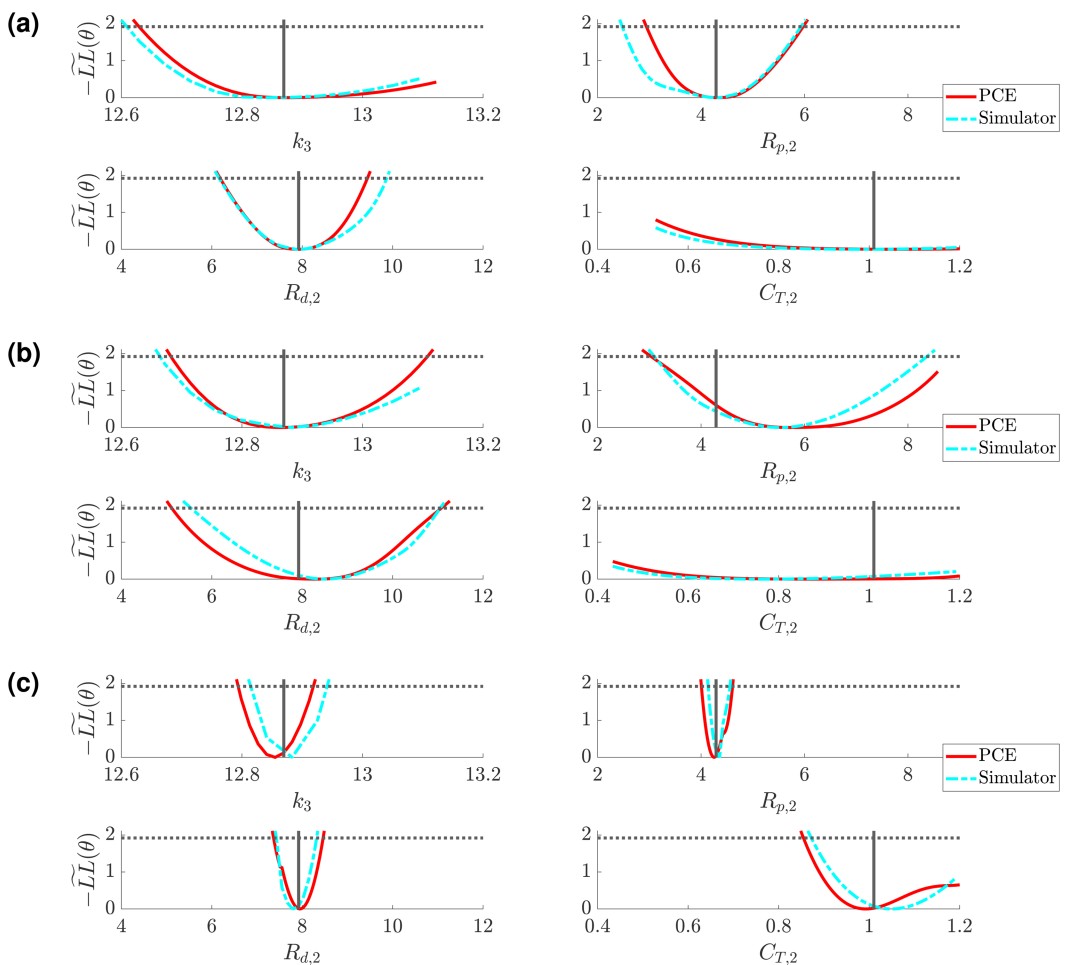

**Fig 12. Comparison between the spectral surrogate (red) and simulator evaluated (cyan) profile-likelihood confidence intervals for the Windkessel boundary conditions using the free parameter set including $k_3$, $R_{p,2}$, $R_{d,2}$, and $C_{T,2}$.** Results are shown for designs $D1$ (a), $D2$ (b), and $D3$ (c).

This approach, though applied to a hemodynamics model, is applicable to broader classes of expensive PDE models, which have yet to formally adopt identifiability methods such as the profile-likelihood. Moreover, our study examines the two most common boundary conditions for hemodynamics models, providing insight for future studies looking to build subject-specific pulmonary models using parameter inference.

## Spectral surrogates

One of our major goals in this article was to illustrate how spectral surrogates and PCEs can be leveraged beyond global sensitivity analysis, requiring no additional simulator evaluations to address parameter identifiability. Here we adopted a similar approach to Nagel et al. [8], which combined PCEs with dimension reduction in the output space via PCA in the context of urban drainage simulation. A similar approach was considered in the study by Paun et al. [22], which compared forward emulation and inverse problem accuracy between PCEs and Gaussian processes for a similar PDE hemodynamics model. Paun et al. also compared full

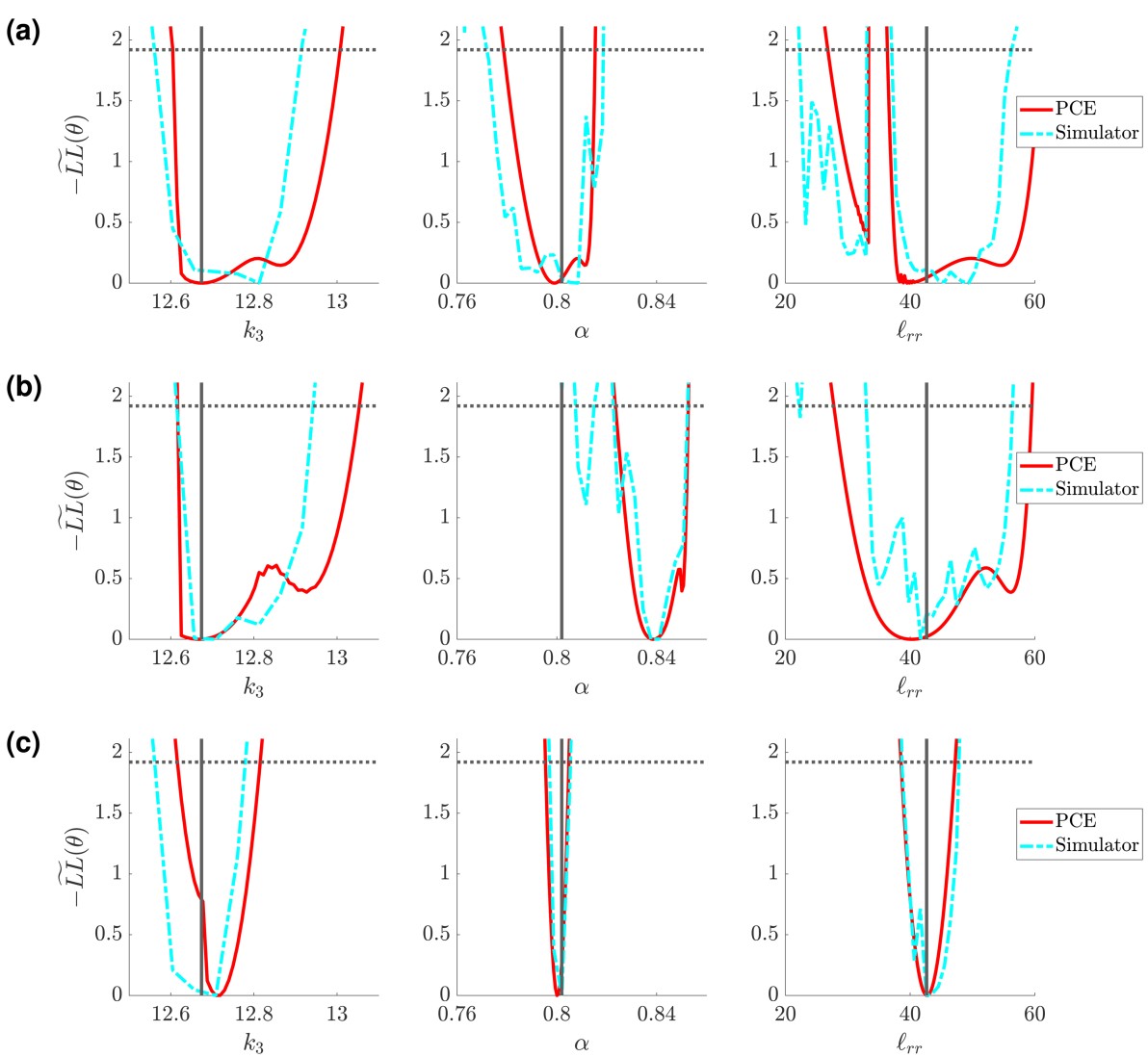

**Fig 13. Comparison between the spectral surrogate (red) and simulator evaluated (cyan) profile-likelihood confidence intervals for the Structured Tree boundary conditions using the free parameter set including $k_3$, $\alpha$, and $\ell rr$.** Results are shown for designs $D1$ (a), $D2$ (b), and $D3$ (c).

dimensional output emulation (i.e., for full time series) to a PCA reduced output, showing similar accuracy between full and PCA reduced outputs.

The PCE surrogate provides similar accuracy across the five different outputs as shown in Fig 4 for the Windkessel boundary conditions. There is a larger emulation error for the Structured Tree model, especially for the pressure output, with outliers on the order of 4% to 15%. Additional variance and error metrics in Figs A, B, C, D, E in S1 File also support the claim that the Structured Tree boundary conditions provide a more variable simulator. The relative error between emulator predictions and test data show that pressure and area emulation is more difficult in the Structured Tree than for flow emulation. In addition, relative errors are much lower for the Windkessel model. This is attributed in part to the relatively larger variance achieved by the Structured Tree model when conducting sampling, as documented

previously [22] and shown in Fig A in S1 File. The variances attributed to each principal component, shown in Table 1, show that five principal components typically capture 99% of the variance. The exceptions include flow in the RPA (vessel 3) for the Windkessel boundary conditions, as well as the two daughter flows in the Structured Tree model. However, the emulation accuracy for the flows across both models are relatively well captured, suggesting that even less than 99% of the variance is necessary for building a sufficiently accurate spectral surrogate.

## Sensitivity analysis: Windkessel model

Variance based sensitivity analysis is considered the gold standard for global sensitivity analysis [4,8], with numerous hemodynamic modeling studies employing Sobol' indices for uncertainty quantification [9–11,33,34]. The use of dimension reduction to quantify the sensitivity of entire time series signals has been employed previously [8], but, to the author's knowledge, has yet to be used for hemodynamics modeling. The study by Huberts et al. [10] used PCEs to identify the sensitivity of a 1D systemic hemodynamics model to vascular stiffness and Windkessel parameters among other parameters. They found that Windkessel resistances were typically influential for both mean flow and systolic radial artery pressure. A similar study by Donders et al. [33] combined Morris screening (another sensitivity method) with Sobol' indices to quantify the impacts of model parameters on mean brachial flow and distal arterial systolic pressure. They found again that Windkessel resistance was influential, whereas compliance was only influential in certain sections of the circulation. The study by Melis et al. [34] used Gaussian processes to emulate systolic and diastolic pressure in a systemic vascular network model, and again found that Sobol' indices were largest for resistance and, in some sections of the systemic vasculature, stiffness. These studies focused on a single quantity of interest for each sensitivity conclusion.

These previous findings are consistent with our results in Fig 5(a), 5(c), 5(e), and 6, which show that compliances are on average the least influential parameters after the exponential stiffness terms $k_1$ and $k_2$. Similar findings came from the prior study by Colebank et al. [24] which found again that parameters related to compliance were the least influential on pulmonary artery pressure conditions. This is consistent with our pressure sensitivities in Fig 6, while we also identify an increased role for the compliance parameters in the left and right pulmonary flows, as shown in Fig 6(b)–6(c). This implies that flow distribution is based in part on compliance values. We also see a dominance of stiffness $k_3$ in influence on area predictions in all three branches. This is intuitive, as stiffness plays a role in the relative area change and area magnitude given by the pressure-area relationship in Eq (4). The time-series sensitivities in Fig 6 show that the model sensitivity fluctuates with the cardiac cycle. Similar to findings by Colebank et al. [24], the results show that pressure sensitivity to distal resistance decreases during systole and then increases in diastole, while stiffness is more influential during systole. The flow sensitivities follow unique trajectories in all three branches, with the LPA and RPA flow sensitivities suggesting that resistance and compliance elements corresponding to the left and right side are most influential in the respective arteries. We note that the MPA flow is a boundary condition, though the MPA sensitivity is calculated at the midpoint where there are some flow fluctuations.

## Sensitivity analysis: Structured tree model

To date, relatively few studies have conducted a sensitivity analysis on models using the Structured Tree boundary conditions. The study by Perdikaris et al. [35] conducted a pseudo-sensitivity analysis with respect to the minimum radius, finding that large values of $r_{min}$ had

drastic effects on pressure in the basilar and radial artery but relatively small effects on flow magnitude. Olufsen et al. [36] conducted a similar pseudo-sensitivity analysis on both systemic and pulmonary circulation models using the Structured Tree, and saw that Structured Tree parameters linked to $\alpha$ and $\beta$ caused significant changes in MPA pressure shape and magnitude while stiffness had a large effect on MPA pulse-pressure. The study by Taylor-Lapole et al. [37] used Morris screening on 1D hemodynamics models for subjects with congenital heart defects, and found that $k_3$ was the most influential on a combination of pressure and flow outputs, with $\alpha$, $\beta$, and $\ell_{rr}$ showing slightly smaller influence. The studies by Paun et al. [22] and Colebank and Chesler [11] use Sobol' indices to rank parameters in arterial and arterio-venous Structured Tree models. They also find that $\alpha$, $\beta$, $\ell_{rr}$ and $r_{min}$ are most influential on pressure and flow, but find mixed results for the sensitivity to stiffness.

Our findings in Fig 5(b,d,f) shows that $\alpha$, $\beta$, $\ell_{rr}$ and $r_{min}$ are most influential on the first principal component of pressure, with $k_3$ being more infleutial on higher components. The flow and area outputs are more influential to stiffness, and thus supports the previous finding by Taylor-Lapole [37] which found stiffness most influential for a combination of mostly flow outputs and some pressure outputs. Our time-dependent Sobol' values in Fig 7 show again that sensitivity magnitudes vary with the dynamics of the cardiac cycle. The $S_i$ and $S_{T_i}$ values for the pressure and area do not qualitatively vary much with the cardiac cycle. In contrast, the flow sensitivity clearly fluctuates with cardiac cycle across the MPA, LPA, and RPA. We focus attention on the LPA and RPA given the use of data as an MPA flow boundary condition. For the LPA (Fig 7(b)), we see that $\alpha$ and $\beta$ are most influential via $S_i$ during the upstroke in systole, with $k_3$, $\ell_{rr}$, and $r_{min}$ becoming more influential in systole. Interestingly, the values of $S_{T,i}$ spike during the flow upstroke without large increases in $S_i$, suggesting that there are a high level of parameter interactions during flow upstroke. In the RPA, there is a large increase in $S_i$ for the parameter $k_3$ during the upstroke of systole, with a similar increase in $S_{T_i}$ as well, suggesting the dominance of this parameter in RPA flow upstroke. The RPA also shows a delayed sensitivity for $\ell_{rr}$, which peaks later than in the LPA. The RPA is longer than the LPA, hence this different in $\ell_{rr}$ may be due to wave propagation and reflection differences due to vessel length. Overall, the findings from Figs 5(b), 5(d), 5(f), and 7 suggest that the Structured Tree parameters are most influential on average, with relatively less sensitivity to $k_3$ than the Windkessel boundary condition results. This suggests that the Structured Tree parameters should all be considered for inference, possibly with $k_3$, while again $k_1$ and $k_2$ should be considered fixed.

## Experimental designs and profile-likelihood

Sensitivity analysis can provide information about parameter influence but generally cannot determine whether parameters are identifiable. The only exception is when parameters are considered functionally non-influential on an output [5]. Sensitivity based fixing of non-influential parameters is typically problem and user dependent [7,10,24] with cut-off values for model sensitivity difficult to determine. Here we leveraged the use of univariate profile-likelihood confidence intervals, which allow for more formal investigations of parameter identifiability. While considered the gold standard for practical identifiability [12,13,18], these methods are computationally expensive and rarely used with PDE models [15,17]. Some notable examples of this include the study by Boiger et al. [20], who considered an integration-based approach for the profile likelihood calculation for inverse problems with PDE constraints. The authors considered a Lagrangian formulation from which they could leverage adjoints for obtaining the profile-likelihoods. The study by Simpson et al. [21] compared the profile-likelihood with Bayesian inference in the context of spatio-temporal

cell movement. The authors found that parameters of a reaction-diffusion like model were practically non-identifiable when diffusion parameters were cell population specific.

Computing the solution to the PDEs using Windkessel or Structured Tree boundary conditions take roughly 3 and 6 seconds, respectively. Non-identifiable parameter sets may require 300 to more than 2000 function evaluations per optimization, and this must be repeated upwards of 100-200 times for each profiled parameter. This is on the order of days or weeks to provide all univariate profile-likelihoods for the full parameter set with a single experimental design. In contrast, generated training data for PCEs can be done in parallel, expediting this process substantially, and the evaluation of the PCEs themselves take 0.1 and 0.05 seconds for the Windkessel and Structured Tree surrogates, respectively. We thus leverage the use of spectral surrogates to provide both sensitivity and identifiability metrics for our PDE model, and further inspect how different experimental designs may lead to different parameter identifiability. The designs chosen reflect possible clinical measurements for the management of pulmonary vascular disease, such as pulmonary hypertension [13,25], and provide insight into how changing the available data may impact the inference of parameters.

Our results in Fig 8(a) support the notion that $k_1$ and $k_2$ are functionally non-identifiable for the Windkessel boundary conditions. Even with the most complex design $D3$, these two parameters show minimal effects on the likelihood function and thus make them impractical to infer. After fixing $k_1$ and $k_2$, we still find that the compliance parameter $C_{T,2}$ is not practically identifiable within the prescribed parameter bounds in Fig 8(b); however, we anticipate that increasing the upper bound will lead to this parameter having finite confidence bounds. Thus, if larger values of $C_{T,i}$ are appropriate for a given problem, we consider the parameter set excluding $k_1$ and $k_2$ to be identifiable when using pressure and flow data (i.e., $D3$). We also considered fixing one set of Windkessel parameters (in the LPA) and inferring the parameters of the RPA to see if there were inherent parameter dependencies, and Fig 8(c) does show that designs $D1$ and $D2$ are more informative when LPA parameters are fixed. This suggests that the LPA and RPA parameters interact with each other and cause issues with identifiability, whereas inferring one set of Windkessel parameters reduces these identifiability issues. This is consistent with findings in prior studies by Colebank et al. [24], Qureshi et al. [38], and Paun et al. [25], which used a reduce set of Windkessel scaling factors instead of the full set of resistance and compliance elements to overcome identifiability issues.

Evaluations of the model along the univariate profile-likelihood, shown in Fig 9, provide a distinction between non-influential and non-identifiable parameters. First, the low sensitivity of $k_1$ provided in Fig 5 implies this parameter is non-influential. Thus, profiling this parameter has little to no effect on pressure (or area, with minimal effects on flow) and thus the output signals lay on top of each other. This corresponds to non-identifiability driven by the small influence of the parameter. Area and flow predictions also overlay for designs $D2$ and $D3$, respectively, reflecting the effects of non-identifiable parameters on output signals. The sensitivity results for $R_{d,1}$ show that it is highly influential, and thus affects pressure, area, and flow predictions. However, $R_{d,1}$ interacts with other parameters of the system (e.g., $R_{p,1}$ and $R_{d,2}$), especially in its effects on pressure and area. This is why the profile-likelihood for $R_{d,1}$ indicates non-identifiability using $D1$ and $D2$. Once flow data is included in $D3$, $R_{d,1}$ becomes identifiable. This is due in part to the importance of resistance on controlling flow distribution between the two branches [22,38]. Finally, sensitivity results show that $k_3$ is influential on pressure, flow, and area. Previous studies have also found that $k_3$ is pivotal in controlling the time-series dynamics of pressure [24,25], hence it is uniquely identifiable across all three designs given that pressure data is consistently used.

The analysis for the Structured Tree model in Fig 10 is qualitatively distinct from the Windkessel results in Fig 8. In particular, the full parameter set (Fig 10(a)) of the Structured

Tree model has numerous local minima and appears riddled with identifiability issues. It has been documented elsewhere [11,25,37] that the stiffness and Structured Tree parameters are interdependent on the simulated hemodynamics, and hence cause issues when calculating the profile-likelihood. To proceed, we use the low sensitivity of $k_1$ and $k_2$ as a starting point for parameter fixing in Fig 10(b). The univariate profile-likelihood confidence intervals clearly improve for several parameters, especially when using $D3$, but still show identifiability issues. In particular, $\alpha$ and $\beta$ appear to oscillate between minima, while $r_{min}$ appears non-identifiable even for $D3$. These results collectively lead to Fig 10(c) and, after seeing that $\alpha$ and $\beta$ are not jointly identifiable, lead to a final three dimensional parameter space including $k_3$, $\alpha$, and $\ell_{rr}$ in Fig 10(d). This is a similar parameter set as those determined by Taylor-Lapole et al. [37] and Paun et al. [22], which used sensitivity analysis to reduce the Structured Tree model dimensionality. We again see that $D3$ by far is the most informative design, and provides evidence that flow data to the left and right pulmonary trunk are informative in inferring model parameters. Model evaluations along the univariate profile-likelihood in Fig 11 support this conclusion as well, again illustrating how data constraints can cause parameter combinations that greatly affect the likelihood, output signals, and parameter identifiability.

Given that the results are contingent on the use of spectral surrogates, we evaluated the true PDE simulation along the univariate profile-likelihoods to ensure that the likelihood geometry and conclusions regarding identifiability are not biased on the surrogate. The results for the Windkessel boundary conditions in Fig 12 show that, across the three experimental designs, the surrogate predicted likelihood and confidence intervals are consistent with simulator derived values. We note that in $D2$, there is a clear bias in the minimum for $R_{p,2}$, though the shape and confidence bounds for the parameter are similar. The overlap between simulator and PCE derived confidence intervals provide evidence that likelihood evaluations by the PCE are consistent. In the case of the Structured Tree boundary conditions, we again see that there are multiple minima in the likelihood landscape, some of which are not identified by the PCE surrogate. For $D1$ in Fig 13, e.g., we see that the univariate profile-likelihood for $\ell_{rr}$ has a rougher landscape than that predicted by the PCE. However, both surrogate and simulator confidence intervals suggest that $\ell_{rr}$ is identifiable. It is clear that the Structured Tree boundary conditions are harder to infer from data, and thus require sufficient amounts of data (e.g., flow data in $D3$) to ensure identifiability.

We see that, in general, $D2$ is only minimally informative over $D1$ in terms of univariate profile-likelihood results. We attribute this finding to the strong coupling between pressure and area as given by Eq (4). Pressure and area signals are nearly identical in shape in all three branches, as shown Fig 3, though the area magnitudes are unique for the three branches. The sensitivity results in Figs 6 and 7 also show that pressure and area have similar dynamic sensitivities and top parameters influencing their dynamics. This is in contrast to flow, used in $D3$, which is distinct from pressure and area, and coupled through more complex mechanisms in the PDE equations themselves. In a similar light, we also see that increasing the available data is not guaranteed to decrease the parameter confidence bounds, as shown in Fig 8(c). Though design $D2$ provides area data, the confidence bounds for $R_{p,2}$, $R_{d,2}$, and $C_{T,2}$ appear to widen relative to $D1$. We attribute this finding with the necessary changes in the likelihood function used in Eq (28). Including additional non-dimensionalized data in the likelihood shifts the calculation of the confidence intervals from pressure-only to a weighted combination of pressure and area (or flow) data. Hence, the profile-likelihood widths may need to be compared with this in mind.

We conclude that, with respect to spectral surrogates, the proposed framework provides insight beyond sensitivity analysis alone for parameter fixing. In particular, employing univariate profile-likelihood based confidence intervals provide a more robust manner in which

parameters are prioritized for inference versus fixing. In addition, our results suggest that flow data is more informative than area data, and provides the likelihood with structure that favors stronger identifiability among parameters. Thus, if feasible, $D3$ is appropriate for future studies that use clinical data for determining subject-specific parameters from either set of boundary conditions.

## Limitations

There are several limitations to the proposed methods and analysis presented here. We use PCEs as a spectral surrogate for the PDE simulator, which are commonly used in engineering [7,8] and biomedical [9–11] applications. However, other emulation strategies, such as Gaussian processes [19,22,32] may provide more accurate surrogate models. Future studies leveraging Gaussian processes or neural networks with the profile-likelihood analysis are warranted. We used dimension reduction for each output quantity to provide a more efficient emulation framework. The PCEs are built on the PCA representation for each quantity of interest, yet there is an inherent correlation structure across outputs as well. This can be overcome by considering multioutput emulation where the covariance structed between outputs is accounted for [32,41]. There is inherent approximation error induced by dimension reduction, which affects emulation accuracy on the true time-series signals. Thus, future work embedding this dimension reduction discrepancy into the solution to an inverse problem should be investigated [32]. We construct univariate profile-likelihood confidence intervals in the absence of measurement noise, instead using the inherent error between the simulator and surrogate as our measurement variance. We only consider univariate profile-likelihood calculations, yet more conclusive multivariate profile-likelihood methods should be considered in the future. For instance, profile-wise analysis as described by Simpson et al. [14] is a necessary next step with this approach. Future applications of this approach with real data will require additional error structure, including a separation between surrogate error, simulator discrepancy, and measurement error [14,25]. There is also a need for more robust methods for updating the likelihood function based on new experimental data. This includes more statistically informed methods for scaling multiple entries in the likelihood function. Nevertheless, this proof of concept study provides evidence that spectral surrogates can be leveraged beyond sensitivity analysis for formal parameter identifiability analysis. Moving forward, we will apply similar methods to PDEs with higher spatial-dimensionality [40]. Finally, we examine three experimental designs for inference in our pulmonary hemodynamics model, which align with possible experimental or clinical measurement protocols [27,28,30]. We consider these designs strictly in terms of parameter identifiability, though there are additional criterion (e.g., optimal experimental design [40]). These approaches should be used in parallel with parameter identifiability analysis.

## Conclusions

This work combines spectral surrogates with the profile-likelihood confidence interval approach to determine model sensitivity and parameter identifiability. We provide evidence that PCEs can be used beyond sensitivity analysis, providing an approach that assesses model sensitivity via Sobol' indices and then further exploits the use of PCEs as a surrogate for constructing univariate profile-likelihood confidence intervals. We assess vector output sensitivity through dimension reduction and provide both PCA-based and pointwise-in-time global sensitivities for the hemodynamics model. Our results show that parameter sensitivity can be used in combination with formal identifiability analyses for a more robust parameter fixing

approach. By using PCEs, we drastically reduce the computation time required for the profile-likelihood from the order of days to the order of hours. Moreover, we show that identifiability analysis and changes in parameter identifiability with different experimental designs can determine whether inverse problems will have unique parameter solutions. In total, this work provides a pipeline for surrogate based analyses that can be used for experimental planning when using complex PDE simulators in biomedical settings.

## Supporting information

**S1 File. Additional results.** Additional numerical results and profile-likelihood analyses for the five test data sets.
(PDF)

## Author contributions

**Conceptualization:** Mitchel J. Colebank.

**Data curation:** Mitchel J. Colebank.

**Formal analysis:** Mitchel J. Colebank.

**Investigation:** Mitchel J. Colebank.

**Methodology:** Mitchel J. Colebank.

**Project administration:** Mitchel J. Colebank.

**Resources:** Mitchel J. Colebank.

**Software:** Mitchel J. Colebank.

**Validation:** Mitchel J. Colebank.

**Visualization:** Mitchel J. Colebank.

**Writing – original draft:** Mitchel J. Colebank.

**Writing – review & editing:** Mitchel J. Colebank.

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
