## [Decision Letter · Decision Letter 0]

8 Jul 2025

PCOMPBIOL-D-25-01134

Assessing parameter identifiability of a hemodynamics PDE model using spectral surrogates and dimension reduction

PLOS Computational Biology

Dear Dr. Colebank,

Thank you for submitting your manuscript to PLOS Computational Biology. After careful consideration, we feel that it has merit but does not fully meet PLOS Computational Biology's publication criteria as it currently stands. Therefore, we invite you to submit a revised version of the manuscript that addresses the points raised during the review process.

Please submit your revised manuscript within 60 days Sep 07 2025 11:59PM. If you will need more time than this to complete your revisions, please reply to this message or contact the journal office at ploscompbiol@plos.org. Please include the following items when submitting your revised manuscript:

We look forward to receiving your revised manuscript.

Kind regards,

Daniele Enrico Schiavazzi, Ph.D.

Academic Editor

PLOS Computational Biology

Mark Alber

Section Editor

PLOS Computational Biology

**Journal Requirements:**

**Reviewers' comments:**

Reviewer's Responses to Questions

**Comments to the Authors:**

Reviewer #1: Please see the attachment.

Reviewer #2: The submission "Assessing parameter identifiability of a hemodynamics PDE model using spectral surrogates and dimension reduction" addresses the important question of parameter estimation, identifiability and sensitivity analysis for a PDE model of blood flow. While the work seems to be of interest and technically correct, the current presentation of the work speaks to a narrow audience (e.g. blood flow modellers) and with some additional care the work could be appreciated by a much broader audience. My comments are intended to help broaden out the appeal of the work since I expect that the findings and the methodology will be of broad interest beyond those computational scientists working in blood flow modelling. I have some comments:

1. The introduction rightly points out that many methods and results for calibration, identifiability analysis and experimental design are known. In addition, working with profile likelihood can be powerful to explore questions of parameter identifiability, yet few studies have attempted this tool for PDE models (i.e. much more has been done for ODE models). References 14-16 are examples where PDE models are studied using PL. The authors could mention https://doi.org/10.1098/rsif.2020.0055 which pre-dates all of the references given in the introduction and the authors certainly use PL to consider identifiability.

2. Throughout the MS (both in the intro and methods), Colebank refers to PL as being for one parameter at a time. This is not incorrect, but it is also worth noting that PL Can be constructed for pairs and triples of parameters so these statements can be slightly adjusted to give a proper definition.

3. The mathematical model, Equations 1-4, seem to be standard within the blood flow modelling community, and the different types of boundary conditions are described appropriately in the Methods section. For me, as a generalist reader, when is missing is some kind of simulation of the model, with various BC choices to show the reader what the solution of the model looks like, and it would be even useful to show solutions with different parameter values so that we can begin to appreciate, visually at least, how the choice of parameter impacts the solution of the model.

4.Part of the challenge I have in reading the paper (and I will not be alone as a generalist reader) is that the first results I see are in Table 1, and Figures 3-4 which summarise some principal components and Sobol indices for those principal components for a set of simulations. Without seeing the simulation data it is impossible for a general reader to understand what the main types of data are, and it is impossible to understand how these Sobol indices relate to questions of parameter estimation, parameter identifiability and experimental design.

5. Figures 8 and 10 present solution of the model "along the profile likelihood" which I take as being a profile-wise prediction. Can you determine the accuracy of these profile-wise predictions by comparing with a full prediction using the full profile log-likelihood? For the moment it is not obvious to me how to interpret the results in Figure 10, for example, where we see that there are relatively small differences between Designs D1, D2 and D3, except we have larger variation with some parameters compared with others. Can the author be clearer about how these prediction intervals are calculated?

6. A key theme in the work is to use a surrogate model instead of dealing with the PDE model. There is nothing wrong with this, but after reading (and re-reading) the article, I remain to be convinced that I understand the accuracy of the surrogate model. This should be demonstrated more thoroughly in the document to provide confidence that the surrogate model is accurately capturing the PDE solution.

**Have the authors made all data and (if applicable) computational code underlying the findings in their manuscript fully available?**

Reviewer #1: None

Reviewer #2: Yes

PLOS authors have the option to publish the peer review history of their article (what does this mean?). If published, this will include your full peer review and any attached files.

Reviewer #1: No

Reviewer #2: No

**Figure resubmission:**
---

## [Decision Letter · Decision Letter 1]

22 Sep 2025

Dear Dr. Colebank,

We are pleased to inform you that your manuscript 'Assessing parameter identifiability of a hemodynamics PDE model using spectral surrogates and dimension reduction' has been provisionally accepted for publication in PLOS Computational Biology.

Best regards,

Daniele Enrico Schiavazzi, Ph.D.

Academic Editor

PLOS Computational Biology

Mark Alber

Section Editor

PLOS Computational Biology

Reviewer #1:

Reviewer #2:

Reviewer's Responses to Questions

**Comments to the Authors:**

Reviewer #1: Please see the attached file.

Reviewer #2: Thank you for the careful revision of the manuscript. I think the work is in much better shape to be understood by a wider audience.

**Have the authors made all data and (if applicable) computational code underlying the findings in their manuscript fully available?**

Reviewer #1: None

Reviewer #2: Yes

PLOS authors have the option to publish the peer review history of their article (what does this mean?). If published, this will include your full peer review and any attached files.

Reviewer #1: No

Reviewer #2: No

---

## [Editor Report · Acceptance letter]

PCOMPBIOL-D-25-01134R1

Assessing parameter identifiability of a hemodynamics PDE model using spectral surrogates and dimension reduction

Dear Dr Colebank,

I am pleased to inform you that your manuscript has been formally accepted for publication in PLOS Computational Biology. Your manuscript is now with our production department and you will be notified of the publication date in due course.

With kind regards,

Anita Estes
